# Structure of the intact tail machine of *Anabaena* myophage A-1(L)

Rong-Cheng Yu [1,2], Feng Yang[1,3], Hong-Yan Zhang[1,2], Pu Hou[1,2], Kang Du[1,2], Jie Zhu[1,2], Ning Cui[1], Xudong Xu[4], Yuxing Chen [1,2], Qiong Li [1,2] ✉ & Cong-Zhao Zhou [1,2] ✉

The *Myoviridae* cyanophage A-1(L) specifically infects the model cyanobacteria *Anabaena* sp. PCC 7120. Following our recent report on the capsid structure of A-1(L), here we present the high-resolution cryo-EM structure of its intact tail machine including the neck, tail and attached fibers. Besides the dodecameric portal, the neck contains a canonical hexamer connected to a unique pentadecamer that anchors five extended bead-chain-like neck fibers. The 1045-Å-long contractile tail is composed of a helical bundle of tape measure proteins surrounded by a layer of tube proteins and a layer of sheath proteins, ended with a five-component baseplate. The six long and six short tail fibers are folded back pairwise, each with one end anchoring to the baseplate and the distal end pointing to the capsid. Structural analysis combined with biochemical assays further enable us to identify the dual hydrolytic activities of the baseplate hub, in addition to two host receptor binding domains in the tail fibers. Moreover, the structure of the intact A-1(L) also helps us to reannotate its genome. These findings will facilitate the application of A-1(L) as a chassis cyanophage in synthetic biology.

Phages are the most abundant biological entities on Earth that infect and co-evolve with the host bacteria[1]. They are largely diverse in morphology and genomic organization, but all tailed phages consist of an icosahedral or prolate capsid encapsulating a dsDNA genome, followed by a complicated tail machine[2]. Besides contributing to horizontal gene transfer in nature, phages have been used as powerful tools in biotechnology, such as genome editing, phage display, biomarkers for bacterial detection, and phage therapy to kill the multidrug resistant pathogenic bacteria[3]. However, most phages possess very narrow host ranges, which keep on evolving along with the infection and amplification. The broad applications of phages are largely restricted by the difficulties to isolate efficient and stable phages against the highly polymorphic host bacteria and to produce commercialized phage agents in large scale.

The host specificity of phage is usually determined by its tail machine[4], especially the receptor binding proteins (RBPs) that constitutes a part of the tail fiber and/or tailspike. At the initiation stage of phage infection, the RBPs recognize and bind to various host receptors exposed on the cell surface, such as lipopolysaccharides and outer membrane proteins[4]. Thus, either construction of chimeric RBPs or introducing mutations at the receptor binding sites of RBPs will enable us to modulate phage host range[5]. Via exchanging domains of heterologous RBPs, Dunne et al. generated dozens of chimeric phages that predictably target the extended *Listeria* serovars[6]. Using a high-throughput method to mutate the tail fiber of T3, Yehl et al. obtained various phage libraries with altered host range[7]. However, these strategies largely depend on the structural information of the complicated and highly diverse tail machine.

[1]School of Life Sciences, Division of Life Sciences and Medicine, University of Science and Technology of China, Hefei, China. [2]Biomedical Sciences and Health Laboratory of Anhui Province, University of Science and Technology of China, Hefei, China. [3]Research Center for Intelligent Computing Platforms, Zhejiang Lab, Hangzhou, China. [4]State Key Laboratory of Freshwater Ecology and Biotechnology, Institute of Hydrobiology, Chinese Academy of Sciences, Wuhan, China. ✉e-mail: liqiong@ustc.edu.cn; zcz@ustc.edu.cn

The cyanophages, which specifically infect cyanobacteria, also possess very narrow host ranges. They are involved in regulating the abundance and population density of cyanobacteria, thus playing a key role in cyanobacterial community succession[8]. A-1(L) is a *Myoviridae* cyanophage isolated in 1970s[9] that specifically infects the model cyanobacterium *Anabaena* sp. PCC 7120. It possesses a 68,304-bp genome of 97 putative open reading frames (ORFs), of which only 20 have been preliminarily annotated[10]. Thus, more investigations on A-1(L) are needed for its application as a chassis cyanophage for developing the environment-friendly agent to control the harmful blooms via lysis of multiple genera of cyanobacteria.

Recently, we reported the capsid structure of A-1(L), which adopts a noncovalent chainmail capsomer construction[11]. Here, we solve the high-resolution structure of A-1(L) tail machine, and reveal a symmetry-mismatched neck anchoring five neck fibers, a long contractile tail with a five-component baseplate, and six pairs of tail fibers. In combination with biochemical assays, we model a bead-chain-like structure of the neck fibers that extend from the unique neck/gp5 pentadecamer, and identify multiple hydrolytic and binding activities of the structural components of the tail machine. Notably, both the long and short tail

fibers are folded back pairwise, with the distal end pointing to the capsid. These structures also enable us to revise and reannotate the genome of A-1(L), and better understand the architecture of myophages and their interplay with the host.

## Results

### Overall structure of A-1(L) tripartite tail machine

Besides the major capsid protein gp4 that constitutes the A-1(L) capsid[11], the results of mass spectrometry showed that there exist 20 extra structural proteins (Supplementary Table 1). Genomic analysis further revealed that the structural genes of A-1(L) are not sequentially clustered in successive operons, but interrupted by distinct genes (Fig. 1a). For example, *gp80-gp82* genes, which might encode the neck fiber, locate far away from other structural genes (Fig. 1a), indicating that these three genes are independently regulated from other structural genes.

Using cryo-electron microscopy (cryo-EM), we solved the structure of A-1(L) tail machine, which consists of three parts: a 205-Å-long neck, a 1045-Å-long contractile tail with a complicated baseplate, and six pairs of tail fibers (Fig. 1b). Totally, 17 of 20 structural proteins could

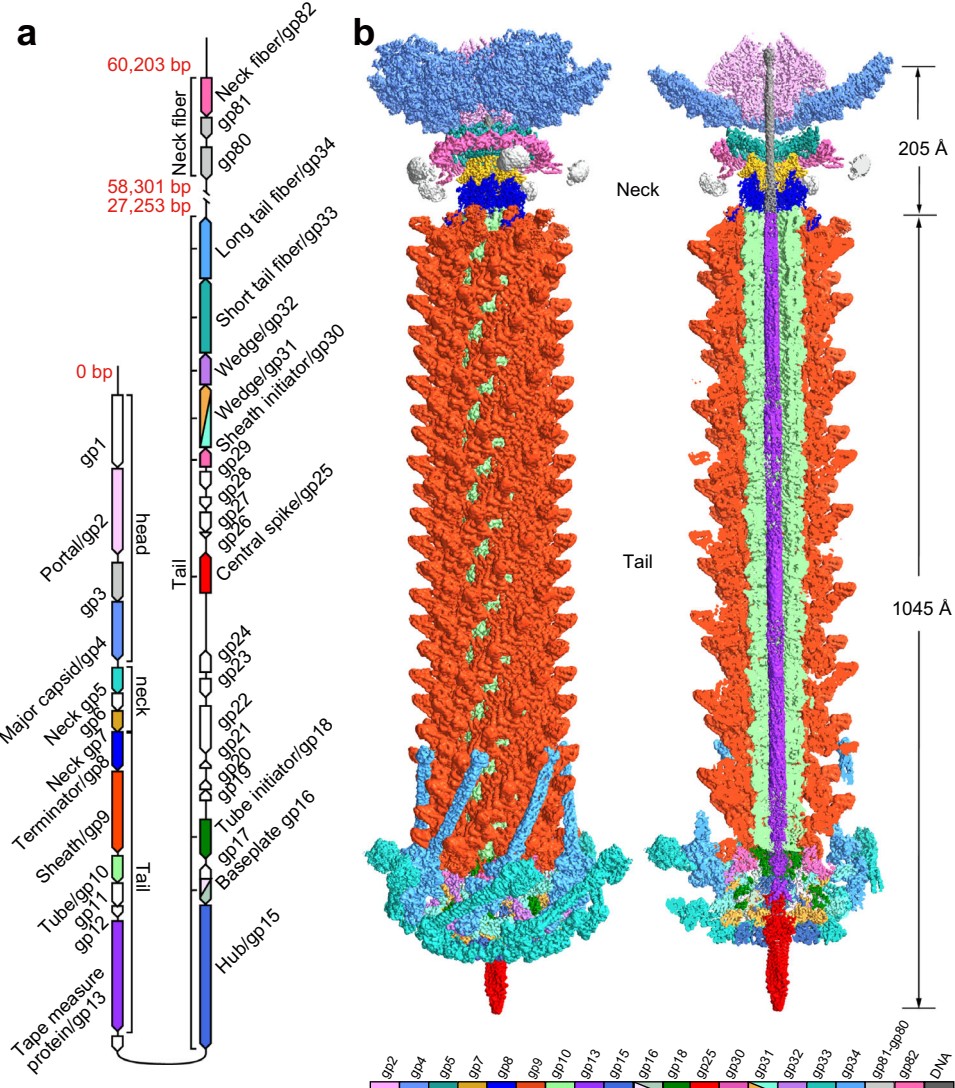

**Fig. 1 | Overall architecture of A-1(L) tripartite tail machine. a** A schematic diagram of the organization of A-1(L) structural genes. The genes in white encode the non-structural proteins that are not presented in the mature A-1(L) particle. The genes in gray encode structural proteins that are identified by mass spectrometry, but not built atomic models in the present structure. **b** The overall cryo-EM map of A-1(L) tripartite tail machine. The structural proteins are colored the same as their encoding genes. The same color scheme is used throughout the manuscript. The lengths of the neck and tail are shown in Å.

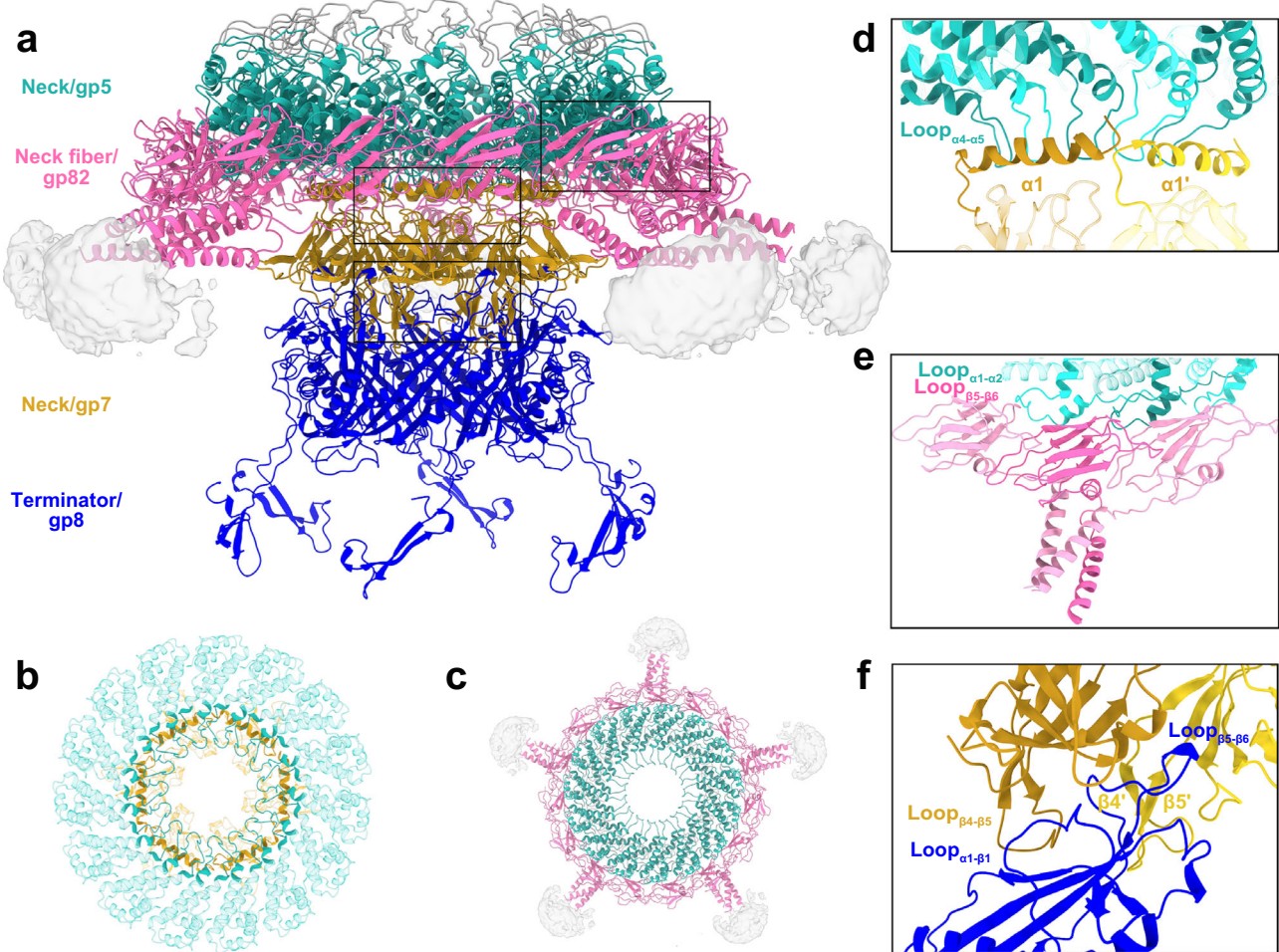

**Fig. 2 | Structure and interfaces of the symmetry-mismatched neck. a** Cartoon representation of the neck, containing the pentadecameric gp5 (green), the hexameric gp7 (gold), and five trimeric gp82N of neck fibers (pink), except for the dodecameric portal. The terminator gp8, which is responsible for the docking of tail to the neck, is also shown in blue. The gray densities indicate the unmodelled parts of the neck fibers. The gp5-gp7 (**b**) and gp5-gp82N (**c**) complexes seen from the head., Paired interfaces of gp5-gp7 (**d**), gp5-gp82N (**e**), and gp7-gp8 (**f**) in magnified views. The secondary structure elements involved in the interactions are labeled. The residue numbers are as follows: Loop$_{\alpha4-\alpha5}$ (residues Ile83-Tyr99) and Loop$_{\alpha1-\alpha2}$ (residues Leu20-Glu30) of gp5; α1 (residues Met1-Leu25), Loop$_{\beta4-\beta5}$ (residues Gly55-Thr65) and β4'-β5' (residues Asn52-Gly55, Ile66-Ala68) of gp7; Loop$_{\beta5-\beta6}$ of gp82N (residues Thr70-Lys75); Loop$_{\alpha1-\beta1}$ (residues Ile15-Val34) and Loop$_{\beta5-\beta6}$ (residues Leu135-Trp152) of gp8.

be modeled in the final density map, except for the scaffolding protein gp3, and the components gp80 and gp81 of putative neck fiber.

The dodecameric gp2 portal, the pentadecameric gp5 and the hexameric gp7 sequentially constitute the neck of A-1(L), to which five neck fibers are attached (Fig. 1b). Each neck fiber comprises a trimeric gp82 with the N-terminal domain clearly modeled, followed by the highly flexible components gp80 and gp81 (Fig. 1b). The dodecameric portal/gp2 forms a channel for the passage of genomic DNA, whereas the hexameric neck/gp7 provides an interface for the docking of tail. The long and contractile tail of A-1(L) contains 24 rings of gp10 hexamers that form the tube, surrounded by the sheath of 24 helically stacked gp9 hexamers (Fig. 1b). The tube and sheath start from the tube initiator/hexameric gp18 and sheath initiator/hexameric gp30, respectively, but both stop at the same terminator/hexameric gp8 (Fig. 1b). At the distal end of the tail, the five-component baseplate possesses a central spike/trimeric gp25, encircled with the hub/trimeric gp15, which is surrounded by six wedges/heterotrimeric (gp31)$_2$-gp32 and further stabilized by six gp16 monomers (Fig. 1b). Anchoring to the flanking sides of the baseplate, two types of tail fibers: six long tail fibers (LTFs, gp34 trimers) and six short tail fibers (STFs, gp33 trimers), are folded back pairwise, with the distal end pointing to the capsid (Fig. 1b).

## The neck of A-1(L) possesses a unique pentadecameric gp5

At the 5-fold vertex of A-1(L) capsid, the 12-fold portal/gp2 dodecamer, the 15-fold neck/gp5 pentadecamer, and the 6-fold neck/gp7 hexamer are sequentially interlocked to form a symmetry-mismatched neck (Figs. 1b, 2a). Moreover, the trimeric N-terminal domain of gp82 (gp82N, residues Met1-Ser125) of each neck fiber attaches to the gp5, whereas the C-terminal domain of gp82 (gp82C, residues Ser126-Leu241) and beyond of the flexible neck fiber could not be modeled due to the poor EM density (Fig. 2a).

Similar to the previously reported portal proteins[12,13], the portal of A-1(L) also consists of five typical domains: barrel, crown, wing, stem, and clip, of which the barrel domain is absent in some phages (Supplementary Fig. 1a). Twelve gp2 subunits form a dodecameric portal with a central channel of 25 Å in diameter at the narrowest tunnel loop (Supplementary Fig. 2a). Notably, we observed two extra densities: one in the central channel and one around the wing of portal dodecamer, which were assigned to two segments of the dsDNA genome in our model, respectively (Supplementary Fig. 2a). The dodecameric portal is surrounded by five major capsid gp4 hexamers in a 12:5 symmetry-mismatch pattern (Supplementary Fig. 2b). The wing domain (residues Met1-Tyr275) of portal interacts with the A domain (residues Ser178-Asn282, Leu353-Gln365) and P domain (residues Asn22-Arg177,

Leu283-Asp352) of corresponding major capsid subunit, forming a circular cleft that accommodates a segment of the genomic dsDNA (Supplementary Fig. 2c).

Considering that all structure-known homologous gp2-gp5 complex of the phages are 12-fold symmetric[2], we first calculated the initial model of gp5 by applying C12 or C1 symmetry; but failed in obtaining a reasonable three-dimensional (3-D) classification. After multiple rounds of attempt, the structures of gp5 and gp5-gp82N complex were finally solved by imposing C15 and C5 symmetry, respectively (Fig. 2a). Structural analysis showed that the neck/gp5 consists of 15 subunits of different fold compared to those previously solved counterparts (Supplementary Fig. 1b); however, it possesses a same 15-fold symmetry as the recently reported collar sheath protein of *Agrobacterium* phage Milano that crosslinks the tail sheath to the neck[14]. Notably, the interface between the 12-fold gp2 and the 15-fold gp5 possesses a rather poor density map, indicating that the relative orientations between symmetry-mismatched gp2 and gp5 vary in A-1(L) particles.

Besides connecting to the portal, the gp5 pentadecamer also interacts with the gp7 hexamer and neck fibers, respectively (Fig. 2b, c). In detail, the ring formed by Loop$_{\alpha4-\alpha5}$ (residues Ile83-Tyr99) of 15 gp5 subunits is hooked by six long α1 helices (residues Met1-Leu25) at the N-terminus of gp7 (Fig. 2d, Supplementary Figs. 1c, 3a), whereas Loop$_{\alpha1-\alpha2}$ (residues Leu20-Glu30) of three gp5 subunits tightly interact with the Loop$_{\beta5-\beta6}$ (residues Thr70-Lys75) of trimeric gp82N (Fig. 2e and Supplementary Fig. 3b). Moreover, docking of tail to the neck is accomplished via direct interactions: Loop$_{\beta4-\beta5}$ (residues Gly55-Thr65) of one gp7 subunit and antiparallel β4′-β5′ hairpin of neighboring subunit interact with Loop$_{\alpha1-\beta1}$ (residues Ile15-Val34) and Loop$_{\beta5-\beta6}$ (residues Leu135-Trp152) of one terminator/gp8 subunit, respectively (Fig. 2f and Supplementary Fig. 3c). In addition, structural analysis suggested that all pairwise interfaces are complementary in shape and electrostatic potential (Supplementary Fig. 3d), enabling the interlocked assembly of the symmetry-mismatched A-1(L) neck.

## The neck fiber adopts a bead-chain-like structure composed of gp82, gp81, and gp80

In the negative-staining EM images, we observed five neck fibers extending from the junction between capsid and tail machine, with an average length of ~100 nm (Fig. 3a). Despite the structure of gp82N that attaches to the gp5 has been determined, the high flexibility makes it hard to solve the structures of remaining moieties of these neck fibers. According to the result of mass spectrometry, we speculated that gp80 and gp81, which are encoded in a same operon with gp82, are also components of the neck fiber. Therefore, we overexpressed and purified gp82-gp81-gp80 complex, which also forms bead-chain-like fibers in vitro (Fig. 3b, c), similar to the in situ neck fibers. As shown in the electrophoresis gel (Fig. 3b), the recombinant neck fiber consists of gp82 and gp80 in a similar molarity, in addition to multiple copies of gp81. Afterwards, size-exclusion chromatography with multi-angle light scattering (SEC-MALS) analysis revealed that this recombinant fiber has a molecular weight of ~894 kDa (Fig. 3d), which is consistent with the sum of theoretical molecular weights of gp81, gp82, and gp80 trimers at a ratio of ~14:1:1.

Afterwards, we modeled the structures of the trimeric gp82, gp81, and gp80 (Supplementary Fig. 4a), using AlphaFold2[15]. The results showed that gp82 indeed possesses two separate domains gp82N and gp82C, which are attached to gp5 and extended outwards, respectively. Structural comparisons showed that the trimeric gp81 exhibits a globular structure similar to the trimeric gp82C in size (Supplementary Fig. 4b), in agreement with the repetitive beads as shown in the negative-staining EM images of both in situ and recombinant fibers (Fig. 3a, c). In contrast, the gp80 trimer of a relatively larger size (Supplementary Fig. 4b) is likely localized to the most distal of the neck fiber, corresponding to the larger bead at the terminus of each neck fiber in the EM images (Fig. 3c). Statistically analyzing dozens of in situ

neck fibers revealed that each ~100-nm-long fiber contains 16 beads (Fig. 3a), consistent with the estimation of ~14 repetitive gp81 trimers inferred from SEC-MALS.

Accordingly, we modeled a putative structure for the full-length neck fiber (Fig. 3e). First, the N-terminal β-hairpins of gp81 trimer interact with the gp82C trimer. Second, 14 gp81 trimers tandemly connect in a head-to-tail manner. Last, one gp80 trimer anchors to the distal gp81 trimer. This modeled structure was further proved by fitting the atomic models of one gp82C trimer and two gp81 trimers into the 6.07-Å-resolution cryo-EM map of recombinant neck fiber (Supplementary Fig. 4c). Moreover, we found that the purified gp82-gp81 and gp81-gp80 complexes could also form a bead-chain-like structure (Fig. 3f, g, Supplementary Fig. 4d), which suggested that the neck fiber is independently self-assembled in prior of being hooked to the neck via gp82N.

## The structures of the tube and sheath

Besides the baseplate, the 1045-Å-long tail of A-1(L) is composed of one tube initiator/hexameric gp18, one sheath initiator/hexameric gp30, the tube and sheath that consist of helically stacked 24 rings of hexameric gp10 and gp9 in two layers, respectively, surrounding the tape measure protein (TMP) gp13, in addition to one terminator/hexameric gp8, from the distal to proximal of the neck (Fig. 1b). Despite the majority of TMP shows a six-fold helical bundle structure, three distal ends of 20 residues (Pro666-Ala689) could be well modeled in three α-helices that form a tripod structure (Supplementary Fig. 5a).

Encircling the TMP, the tube forms a six-start helix structure with a helical rise of 36.2 Å and a twist of 36.2° (Fig. 1b). Each tube subunit adopts a conserved structure (Supplementary Fig. 6), containing a β-barrel domain and an α helix, in addition to a β-hairpin that protrudes towards the next ring to mediate the inter-ring interaction, thus extending the tube (Supplementary Fig. 5b). Notably, stacking of two rings of the tube is also maintained by the complementary electrostatic potential (Supplementary Fig. 5b). Moreover, the sheath also displays a six-start helix structure similar to the tube (Fig. 1b), with each subunit possessing three globular domains in addition to two folded termini, N-tail and C-tail (Supplementary Fig. 6). The domain II (residues Ala364-Ser460) of one sheath subunit stabilizes the N-tail (residues Met1-Ile27) of one subunit and the C-tail (residues Thr461-Val505) of adjacent subunit in the succeeding ring (Fig. 4a), enabling the extension of sheath. Meanwhile, the helix α10 of sheath domain II lies on the concave surface of tube β-barrel domain (Fig. 4a), further stabilizing the interface between the tube and sheath.

Each subunit of tube initiator is composed of two distinct domains: a β-barrel domain and an α/β domain, similar to that in the bacterial extracellular contractile injection systems (eCISs) (Supplementary Fig. 6). Owing to the structural similarity, the ring formed by the β-barrel domains (residues Ser2-Lys180) of tube initiator is compatible with the first ring of the tube (Supplementary Fig. 5c, d), enabling the extension of tube. Moreover, each subunit of the sheath initiator adopts a fold similar to A-1(L) sheath domain II and the previously reported homologs (Supplementary Figs. 5e and 6), which also interacts with the N-tail of one subunit and the C-tail of adjacent subunit in the first ring of sheath (Fig. 4b), leading to the extension of sheath in a similar inter-ring interaction pattern.

In contrast to the initiation with two separate proteins, the termination of the tube and sheath is fulfilled by a same terminator. As shown in Fig. 4c, each terminator subunit simultaneously interacts with two tube subunits via Loop$_{\beta3-\beta4}$ (residues Thr58-Gln71) and β5 (residues Thr125-Leu135), and one sheath subunit via a protruding β7-β8 hairpin (residues Asn202-Ile240), which disrupts the inter-ring interfaces of the tube and sheath, and eventually terminates the extension of tail. Notably, the A-1(L) terminator is structurally different from the previously reported ones in the tailed phages and eCISs (Supplementary Fig. 6).

 

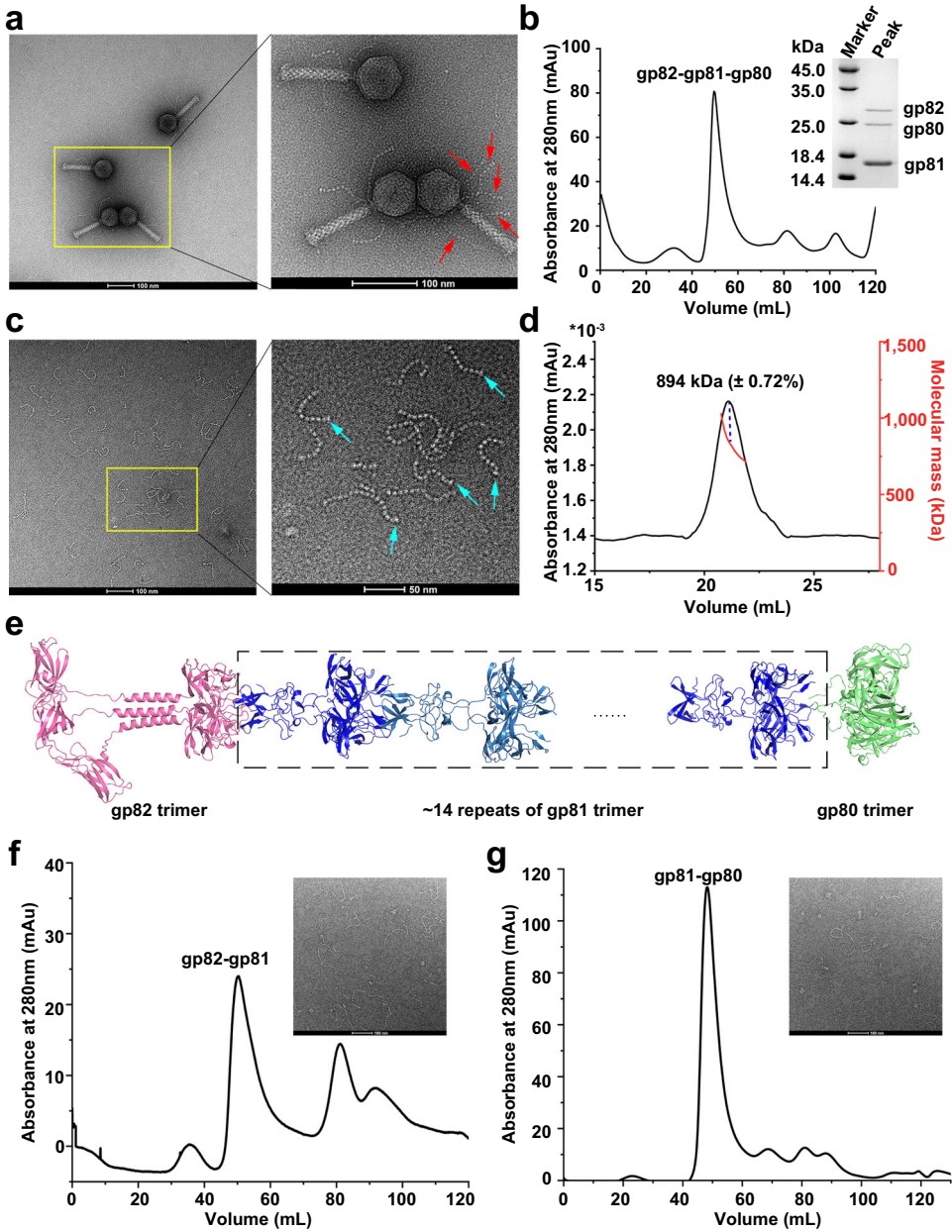

**Fig. 3 | The bead-chain-like structure of the full-length neck fiber. a** The negative-staining EM image of A-1(L) particles. The red arrows indicate the five neck fibers protruding from one A-(L) particle. **b** Gel filtration chromatography profile of the recombinant neck fibers and SDS-PAGE analysis of the highest peak fraction. **c** Negative-staining EM image of the recombinant neck fibers. The cyan arrows indicate the terminal beads with a larger size. **d** SEC-MALS profile of the recombinant neck fibers. The eluted peak corresponds to the X axis. The black line represents the UV absorbance at 280 nm (Y-axis on the left), whereas the jagged short red line indicates the molecular weight (Y-axis on the right). The dashed line indicates the predicted molecular weight of the highest peak fraction. **e** The modeled structure of the full-length neck fiber. The gp80, gp81, and gp82 are colored in green, blue, and pink, respectively. Gel filtration chromatography profiles of the recombinant eGFP-gp82-gp81 (**f**) and gp81-gp80 (**g**) complexes. The corresponding negative-staining EM images of the highest peak fraction were also displayed in the upper right corner. Each experiment in this figure was repeated independently at least three times with similar results.

## The baseplate possesses a hub of dual hydrolytic activities

The central spike/trimeric gp25, hub/trimeric gp15, six monomeric gp16, and six wedges/heterotrimeric (gp31)₂-gp32 form a compact baseplate at the distal end of the tail (Fig. 5a). The central spike locates at the innermost of baseplate, each subunit of which adopts a conserved structure: an N-terminal α-helix, an oligonucleotide/oligosaccharide binding fold, a consecutive β-strand domain and a C-terminal loop (Supplementary Fig. 7a). The N-terminal α-helices (residues Met1-Arg30) of three gp25 subunits form a tripod to interwind with the tripod of three C-terminal helices (residues Pro666-Ala689) of TMP via hydrogen bonds and hydrophobic interactions (Supplementary Fig. 8a).

The two interwound tripods are surrounded by the trimeric hub (Fig. 5a and Supplementary Fig. 8b), each subunit of which adopts an architecture different from the previously reported hub proteins (Supplementary Fig. 7a). Besides the conserved barrel domain that is commonly found in most myophages[16], the hub of A-1(L) possesses three extra domains (Fig. 5b). DALI search[17] combined with structural superposition enabled us to name the three domains as lysozyme (residues Ser56-Ala265), endoglucanase (residues Ser266-Lys430) and peptidase (residues Ser736-Trp899) domain, respectively (Supplementary Fig. 8c). Using in vitro enzymatic activity assays with the commercially available lysozyme as the positive control

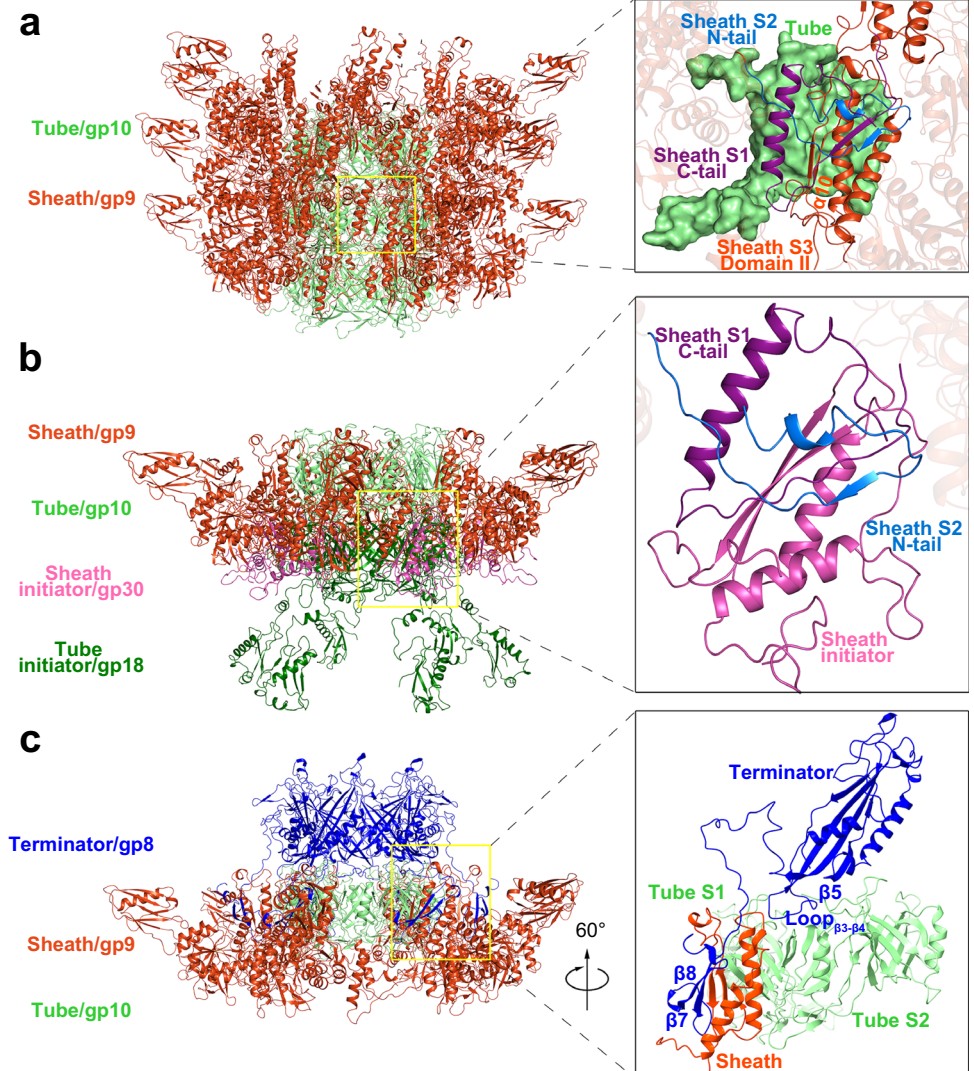

**Fig. 4 | Organization of the sheath and tube.** The cartoon representations showing the interfaces of sheath-tube (**a**), sheath initiator-sheath (**b**) and terminator-sheath/tube (**c**), respectively. The interfaces are also shown in the magnified views at the right. The structural components are colored the same as labeled. The secondary structure elements involved in the interactions are labeled in the insets. The residue numbers are as follows: N-tail (residues Met1-Ile27), domain II (residues Ala364-Ser460) and C-tail (residues Thr461-Val505) of gp9; Loop$_{β3-β4}$ (residues Thr58-Gln71), β5 (residues Thr125-Leu135), β7-β8 (residues Asn202-Ile240) of gp8.

(Supplementary Fig. 8d), we found that the endoglucanase and peptidase domains of hub/gp15 can hydrolyze *Anabaena* sp. PCC 7120 cells (Fig. 5c, d). Moreover, a series of single mutations of conserved catalytic residues (Supplementary Fig. 8e, f), either yield much lower expression level of the recombinant protein or significant decrease of hydrolytic activity (Fig. 5c, d and Supplementary Fig. 8g, h). These results indicated that the hub not only constitutes the structural component of A-1(L), but also participates in the infection.

Six gp16 subunits adopt two conformations (termed gp16A and gp16B), which alternatively anchor on the hub to connect the internal trimeric hub and external hexameric wedge (Fig. 5e, f). The gp16A and gp16B share a same conformation in the N-terminal tail and LysM domain, but differ in the orientation at the C-terminal tail (Supplementary Fig. 9a). Notably, the baseplate gp53 (PDB: 5IV5) of phage T4 also possesses a LysM domain, but fused with different domains (Supplementary Fig. 7b). The C-terminal tail (residues Ile126-Asp152) of gp16A deeply inserts into the barrel domains of hub, whereas that of gp16B forms a four-stranded antiparallel β-sheet with the α/β domain of tube initiator (Fig. 5f, g, Supplementary Fig. 9b, c). Moreover, the N-terminal tail (residues Met1-Leu20) of gp16 runs downwards to recruit the heterotrimeric wedge (Fig. 5f, Supplementary Fig. 9d),

which is composed of two gp31 subunits with different conformations (termed gp31A and gp31B) and one gp32 subunit at the periphery (Supplementary Fig. 9e, f). The N-terminal helical bundle (residues Asn6-Glu81 of gp31 and residues Met1-Asp107 of gp32) of wedge, in addition to two sheath initiator subunits and one tube initiator subunit, encircle the LysM domain (residues Gln36-Ile101) of one gp16 subunit (Fig. 5h, Supplementary Fig. S9g). Notably, compared to those in the previously reported phages and eCISs, the wedge of A-1(L) possesses an insertion domain (Supplementary Fig. 7c). The insertion domain (residues Glu81-Gly180) of gp31A inserts in a cleft on the hub, whereas that of gp31B stretches outside to hold the LTF (Figs. 5f, h, Supplementary Fig. 9h, i). Altogether, all these components form a compact baseplate with a mortise-and-tenon structure.

## The six long and six short tail fibers are folded back pairwise towards the sheath

A-1(L) possesses two types of tail fibers: six LTFs/gp34 trimers (274 Å long) and six STFs/gp33 trimers (258 Å long), all of which anchor to the baseplate in an upward conformation (Fig. 6a, Supplementary Fig. 10a). Each LTF subunit contains a shoulder and an arm domain, whereas each STF subunit consists of four domains (Fig. 6b, c): β-ring,

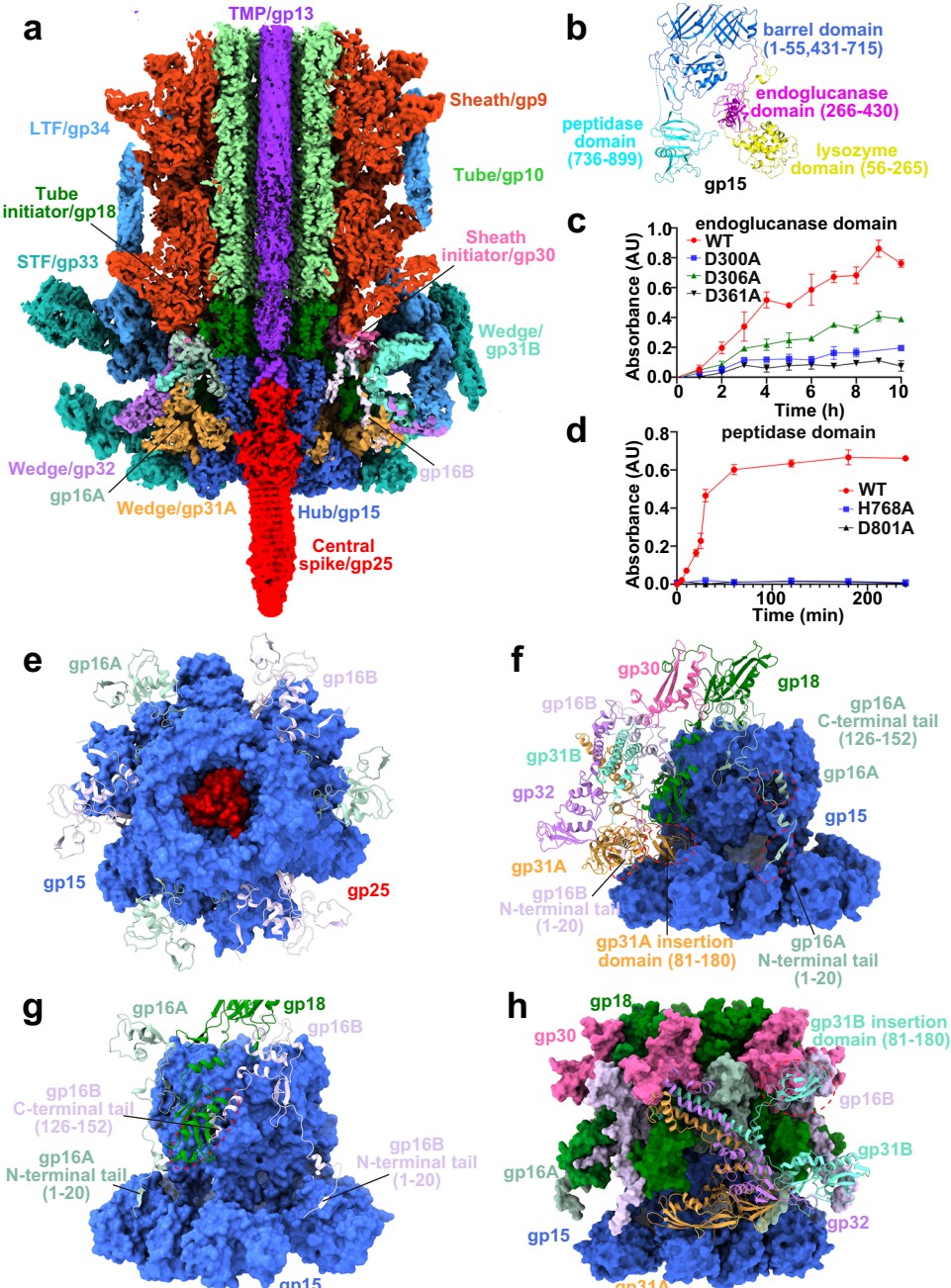

**Fig. 5 | Architecture of the five-component baseplate and a hub of dual hydrolytic activities.** **a** Longitudinal cut view of the baseplate. The structural components of the baseplate in the cryo-EM map are colored the same as labeled. **b** Cartoon presentation of the hub with domains labeled and colored differently. The residue number of each domain is also indicated. Plots of cytochrome released from lytic *Anabaena* sp. PCC 7120 cells upon the treatment of endoglucanase (**c**) and peptidase domain (**d**) of the hub, in addition to their single mutants. The release of cytochrome was detected at the absorbance of 610 nm. Each data point is the average of three independent experiments (*n* = 3), and error bars represent the means ± SD. **e** The central spike complexed with hub and the surrounding six gp16 monomers seen from the head. The central spike-hub complex is shown as the surface, whereas six gp16 monomers are displayed as cartoons. **f** Side view of the wedge in complex with the hub, gp16, tube initiator, and sheath initiator. The hub is shown as the surface, whereas other components are shown as cartoons. **g** Two gp16 subunits with different conformations and one tube initiator subunit anchor on the hub. The hexameric hub is shown as the surface, whereas the gp16 and tube initiator are shown as cartoons. **h** Structure of the baseplate with the tube/sheath initiator, in which the heterotrimeric wedge is shown as cartoons. The N- and C-terminal tails of gp16A/gp16B and the insertion domain of gp31A/gp31B are highlighted with red circles and labeled in (**f**–**h**).

joint, stem and cell wall binding domain (CBD). Although the arm domain of LTF is similar to the tail fiber gp37 of phage T4, the tail fibers of various phages usually adopt different structures (Supplementary Fig. 10a), which enable the recognition and binding to diverse hosts. The shoulder domain (residues Met1-Ile155) of LTF is anchored to the wedge of baseplate, via interacting with the insertion domain (residues Glu81-Gly180) of gp31B and the C-terminal loop (residues Val129-

Glu151) of gp32 (Fig. 6b). In contrast, the 180-Å-long arm domain (residues Pro156-Val379) of LTF is folded back and lies along the groove on the sheath (Fig. 6b). Eighteen β-ring domains (residues Thr2-Gln110) of six trimeric STFs form a ring structure that encircles the distal plane of six wedges, whereas the joint domains (residues Leu111-Gln215) bind to the periphery of six wedges (Fig. 6c), making the STFs tightly anchor to the baseplate. Moreover, the long helical bundle

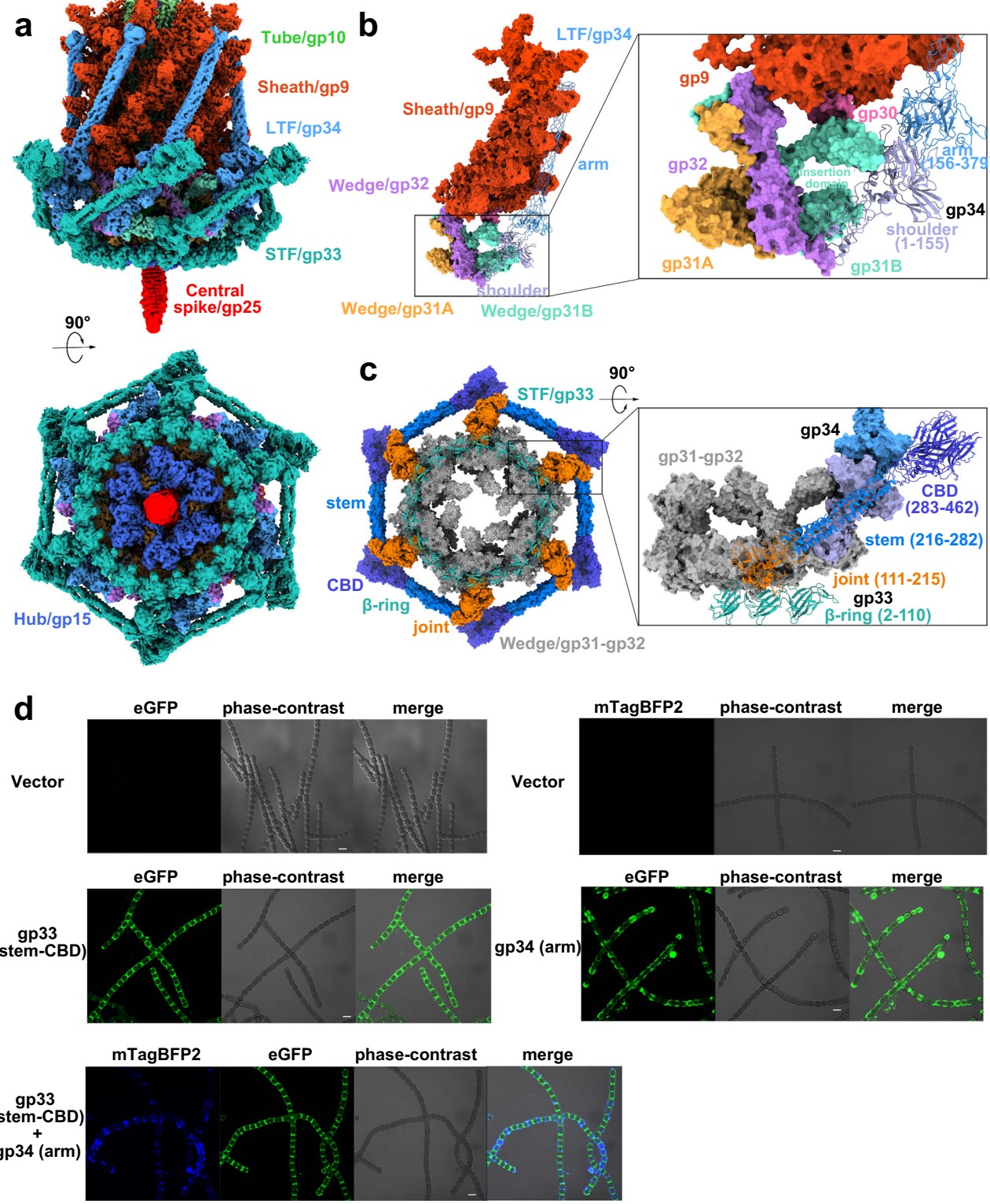

**Fig. 6 | The long and short tail fibers (LTFs and STFs) are folded back pairwise towards the sheath and function as receptor binding proteins (RBPs). a** Side (top) and head-distal (down) views of the baseplate attached with the tail fibers. The structural components in the cryo-EM map are colored the same as labeled. **b** The LTF interacts with the wedge and sheath. The magnified interface is shown as an inset and labeled. **c** Head-distal (left) and side (zoom-in, right) views of the STF attached to the wedge and LTF. The heterotrimeric wedge is colored in gray. The trimeric gp34 and gp33 are shown as cartoons in (**b**–**c**), respectively, with domains colored differently, whereas the remaining components are shown as the surface in each panel. **d** Confocal images of vectors and different constructs of gp33 and gp34 incubating with *Anabaena* sp. PCC 7120 cells. The green fluorescence (eGFP) is excited by the wavelength of 488 nm, whereas that for the blue fluorescence (mTagBFP2) is 405 nm. For the competition binding assays, gp33 (stem-CBD) and gp34 (arm) are fused with eGFP and mTagBFP2, respectively. Magnification, 100×. Scale bar: 5 µm. Each experiment was repeated independently at least three times with similar results.

formed by the stem domains (residues Asn216-Asn282) of STF trimer runs along the groove formed by the trimeric shoulder domain (residues Met1-Ile155) of the LTF (Fig. 6c). The overall folded conformation of these two types of tail fibers suggested a state of A-1(L) in prior of attachment to the surface of host cell.

It is known that the RBP is responsible for triggering phage infection via recognizing and binding to the host[4]. For A-1(L), we previously found that antibodies against either STF or LTF could strongly inhibit the infection, suggesting that they might be essential for the adsorption of A-1(L) to the host cell surface[18]. Structural analysis showed that the arm domain of LTF and the CBD domain of STF most likely possess the binding motifs that recognize the host receptors. Using in vitro binding assays, we observed that the arm domain of LTF, but not the CBD domain of STF, binds to the cell surface of *Anabaena* sp. PCC 7120 (Fig. 6d, Supplementary Figs. 10b, 11a, b). In contrast, the stem-CBD domain of STF could bind to the host cell surface (Fig. 6d, Supplementary Fig. 11a). Further competition binding assays showed that the arm domain of LTF and the stem-CBD domain of STF could simultaneously bind to the surface of host cells (Fig. 6d). Moreover, similar binding activities were observed for the recombinant full-length LTF and STF (Supplementary Figs. 10c, 11c). Altogether, it suggested that the STF/gp33 and LTF/gp34 are indeed RBPs, which recognize and bind to different receptors on the host cell surface.

### Reannotation of A-1(L) genome based on the assigned structural components

During building the atomic models of structural proteins, we found several controversies between our EM density maps and the deposited protein sequences (https://www.ncbi.nlm.nih.gov/nuccore/KU234533.1). Thus we re-sequenced the purified A-1(L) genome and revealed 6 insertions, 3 mutations, and 2 deletions in the 67,884-bp genome (Supplementary Table 2), compared to the previously reported sequence[10]. The insertions of a base T in the coding regions of *gp15*, *gp18*, and *gp31* (corresponding to the previous *gp16*, *gp19* and *gp32*), respectively, enable the coding of correct structural proteins as shown in our structures. The structure of the central spike N-terminal α-helix that interacts with the TMP indicated that it has a longer coding region. A deletion of 423 bp covering the 3′ non-coding region of previous *gp72* and 5′ coding region of *gp73* was verified by PCR experiments. The other variations are most likely due to polymorphism and/or co-evolution.

Furthermore, in combination with BLASTp search[19] against multiple databases and AlphaFold2 prediction[15], we reannotated the genome of A-1(L) with 95 ORFs. Eventually, 57 of 95 ORFs (60%) are functionally annotated, which could be classified into four groups: structural protein, nucleotide metabolism, DNA replication and packaging, and other functions (Supplementary Fig. 12 and Supplementary Table 1). For example, besides the major capsid, tube and sheath, the remaining 18 structural components are all clearly assigned and annotated. Notably, due to the high diversity of most phages, usually less than 40% of the ORFs could be functionally annotated; thus systematic determination of the phage 3-D structure should become a powerful tool to revise and precisely annotate the genome.

## Discussion

Here we solved the cryo-EM structure of the tail machine of freshwater *Myoviridae* cyanophage A-1(L), and clearly elucidated the overall architecture, especially the interactions among different structural components. The dodecameric portal/gp2, gp5 pentadecamer and gp7 hexamer form a symmetry-mismatched neck, which are perfectly accommodated to the 5-fold vertex of icosahedral capsid. Six gp16 proteins of dual conformations fill the gaps between the internal trimeric hub-spike and external hexameric wedge, forming a solid baseplate. Along the helical bundle of TMP, the tube and sheath, in addition to the corresponding initiators and terminator, constitute a two-layered contractile tail. The flexibility of pairwise interfaces of the

neck and tail might facilitate the relative rotation among structural components, in turn enabling the tail contraction and DNA ejection.

The neck fibers of phage T4 were proposed to facilitate the folding back of LTFs during assembly, or to control the contraction of LTFs under unfavorable conditions[20]. However, only low-resolution structures of phages T4 and ΦRSL1 neck fibers have been reported[20,21], which contain 6 or 12 fibritin molecules symmetrically binding to the 12-fold neck. A-1(L) possesses five neck fibers with gp82N tightly attached to the pentadecameric neck/gp5 (Fig. 2a), representing a neck and neck fiber of distinct symmetry. Notably, the neck fiber gp82N adopts a fold similar to the previously reported component of bacterial pilus (Supplementary Fig. 1d). Moreover, we modeled a bead-chain-like structure composed of gp82, gp81, and gp80 for the full-length neck fiber (Fig. 3). The distal carbohydrate-binding module in gp80 of the neck fiber does not possess binding activity toward *Anabaena* sp. PCC 7120 (Supplementary Figs. 10d, 11d), indicating that the neck fiber might not participate in the primary recognition of host cells.

In *Myoviridae* and *Siphoviridae* phages, the tail length is determined by the TMP, the expulsion of which from the tube might trigger the release of genomic DNA[22,23]. However, no structure of full-length TMP at atomic resolution is yet available. Here we solved the structure of three C-terminal 20-residue helices at the distal of A-1(L) TMP (Supplementary Fig. 5a), which is similar to those in *Staphylococcus aureus* phage 80α[24], *Escherichia coli* phages T5[25] and λ (PDB: 8IYL) (Supplementary Fig. 6). These three distal helices of TMP usually form direct interactions with specific baseplate proteins (Supplementary Fig. 6), such as central spike and hub. However, TMP was proposed to adopt a six-fold helical bundle structure in *Myoviridae* cyanophage Pam3[13], which is consistent with our observation for the majority of A-1(L) TMP (Supplementary Fig. 5a). The absence of the other three copies of TMP segment at the distal is probably due to the proteolytic cleavage during tail morphogenesis[22].

The phage T4 harbors a complicated baseplate, with six STFs retracted at the head-distal plane in addition to six LTFs folded back and binding the capsid[26,27]. The results of cryo-electron tomography indicated that T4 first releases LTFs to recognize the host receptors and find an optimal site for infection; then, extends STFs downwards to irreversibly bind to the cell surface, eventually triggering tail contraction and DNA ejection[28]. Here we report the intact structure of the baseplate and tail fibers of A-1(L), all of which are folded back pairwise towards the sheath (Figs. 5, 6). Both the LTFs and STFs function as RBPs to bind to the cell surface of host *Anabaena* sp. PCC 7120, and recognize different host receptors. For example, a previous report showed that the LTF specifically targets the O antigen of lipopolysaccharides[18], whereas the receptor of the STF remains unknown. Notably, we observed in some A-1(L) particles that the tail fibers are released from the groove on the sheath and binding to the co-purified vesicles (Supplementary Fig. 13). However, further investigations, such as cryo-electron tomography, on the infection of A-1(L) against its host *Anabaena* sp. PCC 7120, would help us to reveal which kind of fibers deploy first upon infection.

For myophages, the β-helix of central spike might act as a drill to penetrate the outer membrane of host cell[29,30]. As shown in Supplementary Fig. 7a, the central spike of A-1(L) is structurally similar to the component All3320 (PDB: 7B5H) of the *Anabaena* sp. PCC 7120 eCIS[31]. Upon the penetration of outer membrane of host cell, the hub of A-1(L), which possesses the peptidase and endoglucanase activities, will further hydrolyze the surrounding cell wall and facilitate the injection of genomic DNA.

In sum, we solve the intact structure of the contractile tail machine of myophage A-1(L). Structural analysis combined with biochemical assays reveal the binding and hydrolytic activities of the tail machine towards the host cell, which work together to enable the efficient infection. Moreover, the high-resolution structures of the

structural components largely improve the reannotation of A-1(L) genome. Our present structure of the tail machine and the previously reported capsid structure make A-1(L) an ideal chassis cyanophage for the future applications in synthetic biology.

## Methods

### Purification, genome sequencing and reannotation of A-1(L)

Until grown in BG11-PC medium at 30 °C to an $OD_{750\ nm}$ of 0.8, 3 L of *Anabaena* sp. PCC 7120 cells (kindly provided by Prof. Cheng-Cai Zhang from Institute of Hydrobiology, Chinese Academy of Sciences) were infected by A-1(L) at a multiplicity of infection = 1. Then A-1(L) particles were harvested and resuspended in SM buffer (50 mM Tris-HCl pH 7.5, 10 mM $MgSO_4$, 100 mM NaCl), after prolonged centrifugation and density gradient centrifugation of cell lysates. The target phages were collected by syringe and dialyzed against SM buffer. Negative-stain EM was used to check the purity and integrity of A-1(L) viral particles. In order to identify the structural components, the purified phage particles were applied to 4–12% gradient polyacrylamide gel, which was further analyzed by liquid chromatography-mass spectrometry.

The genomic DNA was extracted using the UNIQ-10 Column Virus Genomics DNA Isolation Kit (Sangon Biotech Shanghai Co., Ltd.) in accordance with the manufacturer's instruction. Afterwards, the extracted DNA was interrupted to construct libraries with different insertions via whole-genome shotgun strategy; and then, paired-end (PE) sequencing was performed on these libraries using Next Generation Sequencing technology, based on the Illumina NovaSeq platform (Nanjing Personal Biotechnology Co., Ltd.). Totally, 7,426,462 reads with an average length of ~150 bp were assembled into one contig with the software SPAdes[32]. The sequencing coverage was approximately 16,369-fold for A-1(L).

The ORFs were predicted by GeneMarkS (http://exon.gatech.edu/GeneMark/genemarks.cgi). Then the translated ORFs were searched against the nr protein database in NCBI using the BLASTp program, the believable results of which should possess $e < -10^{-3}$. The hit with the minimal e-value was regarded as ortholog for each encoded protein. Meanwhile, HHpred (https://toolkit.tuebingen.mpg.de/hhpred) analyses against the PfamA database and conserved domain database were also respectively carried out with the default parameters[33].

### Cryo-EM sample preparation and data collection

3.5 μL of concentrated A-1(L) particles was applied on freshly glow-discharged Quantifoil R2/1 Cu 300 copper grids. The grids were then blotted with filter paper for 3 s and −1 N blotting force, after waiting 20 s for the adsorption. Using a Vitrobot Mark IV (FEI), the grids were plunged into liquid ethane cooled with liquid nitrogen at 8 °C under 100% humidity. Cryo-EM movies (40 frames, each 0.15 s) were collected in counting mode at a nominal magnification of 81,000×, under 300 kV FEI Titan Krios electron microscope equipped with a K3 Summit direct electron camera (Gatan) at University of Science and Technology of China. In total, 6000 movies, with a defocus range of −1.5- −2.5 μm, a total accumulated dose of 50 e⁻/Å², and a final pixel size of 1.07 Å, were recorded using EPU (Thermo Fisher Scientific).

3.5 μL of recombinant neck fibers (gp82-gp81-gp80) were applied on Quantifoil R2/1 Cu 300 mesh grids, which were then blotted with filter paper for 6 s and −1 N blotting force. The grids were also plunged into liquid ethane cooled with liquid nitrogen using a Vitrobot Mark IV (FEI) under 100% humidity at 8 °C. Cryo-EM images were recorded under the parameters the same as those for A-1(L) particles, but in super-resolution mode. A total of 1062 movies (40 frames, each 0.115 s, total dose 55 e⁻/Å²) with a defocus range from −2.5 to −1.8 μm were collected.

### Cryo-EM data processing

For A-1(L), motion correction and dose weighting of movie frames were performed using MotionCor2[34], and the defocus values were determined by CtfFind4[35]. To obtain exact coordinates of the portal as well as other components of the neck, we first performed calculations using I3 symmetry and obtained a 3.47 Å structure of capsid using RELION3.1[36]. Based on the "block-based" reconstruction method[37], we distinguished the portal vertex (-1/12 of the total subparticles) from other 11 pentameric vertices by applying C5 symmetry. Then via imposing the C1 symmetry, we obtained the 4.0 Å-resolution EM map of the portal vertex in complex with capsid. Considering that all structure-known gp2-gp5 complexes are 12-fold symmetric, we calculated the initial model of adapter by applying C12 or C1 symmetry; but failed to obtain a reasonable 3-D classification. After multiple rounds of attempts, the structure of gp5 complexed with neck fiber was finally solved using C5 symmetry. Subsequent rounds of sequential local classification and subparticle reconstruction yielded three EM maps of portal (3.44 Å), gp5-neck fiber (3.66 Å) and gp7-terminator (3.44 Å), using 12-fold symmetry, 5-fold symmetry, and 6-fold symmetry calculations, respectively.

For the helical reconstruction of the tail, 48,042 particles were manually selected. After sorting, polishing and 3D refinement, the EM density of tail with extended conformation was determined to 2.99 Å, with a helical rise of 36.2 Å and a twist of 36.2°. However, due to the flexibility of loops between domain I and domain III, the outside density for the tail sheath is rather poor. Therefore, we alternatively reconstructed the sheath structure using the density from the baseplate, instead of helical symmetry.

For the baseplate region, 73,159 particles were manually selected, and multiple rounds of sequential local classification were performed. Finally, 41,062 good particles were selected for 3D refinement imposed with C3 and C6 symmetry, yielding a final resolution of 3.44 and 3.26 Å, respectively. To further improve the resolution of the CBD domain of STF, we moved coordinates from the center to the end according to the "block-based" reconstruction method[37], reextracted the particles, and performed 3D refinement with C1 symmetry, yielding a final resolution of 3.76 Å.

For recombinant neck fibers, the movie frames were motion corrected and dose weighted using cryoSPARC 3.3.2[38], and binned 2-fold to yield a pixel size of 1.07 Å. In total, 906,481 particles were automatically picked and extracted for 2D processing. Then, 355,765 particles from the best classes were put into ab initio reconstruction, and finally 183,297 particles were used for heterogeneous refinement, yielding a final reconstruction map of 6.07 Å.

The flowcharts of cryo-EM data processing of the A-1(L) virion and the recombinant neck fibers were respectively summarized in Supplementary Figs. 14, 15. All resolutions of the cryo-EM maps (Supplementary Fig. 16) are determined by Golden standard Fourier shell correlation using the 0.143 threshold[39].

### Model building and refinement

The initial models were generated using Alphafold2[15], and automatically fitted into the corresponding maps. Then, the models were manually adjusted using COOT[40], followed by automatic real-space refinement in PHENIX[41]. High quality cryo-EM density maps allowed us to reconstruct atomic models of 17 structural proteins. The final models were evaluated by Molprobity[42]. The cryo-EM parameters, data collection and refinement statistics were summarized in Supplementary Table 3. Structure figures were prepared with Chimera[43], ChimeraX[44], and PyMOL (https://pymol.org/2/). The interfaces were analyzed with PDBsum[45].

### Cloning and plasmids

The genes encoding neck fiber (gp82-gp81-gp80) were amplified from the genomic DNA of A-1(L), and cloned into the pET28a-derived vector with a C-terminal His₆-tag using a ClonExpress® II One Step Cloning Kit (C113-02, Vazyme Biotech co., Ltd). Using this plasmid as a template, the plasmids containing gp82-gp81 with a N-terminal His₆-tag and

gp81-gp80 with a C-terminal His$_6$-tag were respectively constructed. The genes encoding three domains of hub gp15 were synthesized at Sangon Biotech after codon optimization, and then subcloned into the pET19b vector with an N-terminal His$_{10}$-tag, respectively. All single mutants of the hub were constructed by a standard two-step PCR strategy. The genes encoding gp33 and gp34 were amplified from the genomic DNA of A-1(L), and cloned into the pET28a-derived vector with an N-terminal His$_6$-eGFP tag using a ClonExpress® II One Step Cloning Kit (C113-02, Vazyme Biotech co., Ltd). Similarly, the CBD domain of gp33, the stem-CBD domain of gp33, and the arm domain of gp34, all of which were fused with an N-terminal His$_6$-eGFP tag, were respectively constructed. In addition, the gp34 and its arm domain alone were also respectively cloned into the pET28a-derived vector with an N-terminal His$_6$-mTagBFP2 tag. All recombinant plasmids were verified by DNA sequencing (Sangon Biotech Shanghai Co., Ltd.).

### Protein expression and purification

The recombinant plasmids pET28a-gp82-gp81-gp80-His$_6$ and different constructs of gp33 and gp34 were respectively transformed into *E. coli* strain BL21 (DE3), and grown at 37 °C in LB medium containing 30 μg/mL kanamycin until the OD$_{600\ nm}$ reached 0.8. In contrast, the recombinant plasmids pET19b-His$_{10}$-gp15 lysozyme domain, pET19b-His$_{10}$-gp15 endoglucanase domain, pET19b-His$_{10}$-gp15 peptidase domain and their single mutants were respectively transformed into *E. coli* strain BL21 (DE3), and grown at 37 °C in LB medium containing 30 μg/mL ampicillin until the OD$_{600\ nm}$ reached 0.8. Subsequently, protein expression was all induced with 0.2 mM isopropyl β-D-1-thiogalactopyranoside for another 20 h at 16 °C.

Cells expressed neck fiber were harvested by centrifugation at 8000 *g* for 4 min and resuspended in 40 mL binding buffer (20 mM Tris-HCl pH 7.0, 100 mM NaCl, and 7 mM β-mercaptoethanol). The supernatant containing the target protein was transferred to a nickel-nitrilotriacetic acid column (GE Healthcare, Chicago, IL, USA) pre-equilibrated with binding buffer, after 15 min of sonication and 30 min of centrifugation at 12,000 *g*. The target protein was eluted with 400 mM imidazole, and further applied to a HiLoad 16/600 Superdex™ 200 column (GE Healthcare, US) pre-equilibrated with the binding buffer. The His$_6$-gp82-gp81 and gp81-gp80-His$_6$ complexes were expressed and purified in the same way as the neck fiber of gp82-gp81-gp80-His$_6$.

The buffer for different constructs of gp33 and gp34 is 20 mM Tris-HCl pH 7.5, 100 mM NaCl, whereas that for three domains of hub and single mutants is 20 mM Tris-HCl pH 7.5 or 8.5, 100 mM NaCl, 10% glycerol. The protein expression and purification procedures were the same as that described for neck fiber, except that HiLoad 16/600 Superdex™ 75 column (GE Healthcare, US) were used. The gel electrophoresis and electron microscopy were used to assess the purity of proteins.

### Oligomeric state analysis

The oligomeric state of recombinant neck fiber was determined using size exclusion chromatography with multi-angle light scattering (SEC-MALS). Protein samples (1 mg/mL, 100 μL) were eluted through the Superdex 200 Increase 10/300 GL column (GE Healthcare, US) pre-equilibrated with binding buffer, which was connected to an eight-angle MALS detector (DAWN HELEOS II, Wyatt Technology, Santa Barbara, CA, USA) and a refractive index detector (Optilab T-rEx, Wyatt Technology). The results were processed and analyzed using ASTRA 7.0.1 software (Wyatt Technology), and finally plotted using Origin 2022.

### Cell hydrolytic experiments

300 μL of *Anabaena* sp. PCC 7120 cells at exponential phase was incubated at 30 °C with proteins (peptidase domain of the hub and their mutants) at a final concentration of 3 μM for indicated times.

After incubation, cells were centrifugated at 6000 × *g* for 10 min, and the supernatant were further applied to detecting the absorbance at 610 nm using a Beckman DU800 spectrophotometer. The absorbance value corresponds to the amount of released cytochrome after cell lysis. The lytic abilities of endoglucanase/lysozyme domain of the hub and mutants were detected in the same manner as peptidase domain, except for pretreatment of the cells with EDTA to make the outer membrane be permeable. Briefly, 300 μL of cells were collected, washed once with BG11-PC, and incubated for 60 min at 30 °C upon the addition of 1 mM EDTA. After incubation, the cells were centrifuged at 6000 × *g* for 10 min, and washed twice with BG11-PC. Then, the bacterial pellet was resuspended in BG11-PC, and applied to further measurements. A commercially available lysozyme (12650-88-3, Sangon Biotech Shanghai Co., Ltd.) was applied as a positive control. All the experiments were repeated at least three times. The final results were analyzed and plotted using Origin 2022.

### Fluorescence binding assays

300 μL of *Anabaena* sp. PCC 7120 cells at exponential phase was harvested and resuspended in SM buffer (50 mM Tris-HCl pH 7.5, 10 mM MgSO$_4$, 100 mM NaCl) of equal volume. Then, the cells were incubated with 10 μg protein fused with eGFP tag for 1 h at 30 °C. The mixtures were washed three times with SM buffer, and observed via a laser-scanning confocal microscope (ZEISS LSM710 and LSM880) using a 100× objective with oil immersion and phase contrast. For the competition binding assays, the host cells were incubated with differently labeled tail fibers in a same amount: gp33/gp33 (stem-CBD) with eGFP tag (green), and gp34/gp34 (arm) with mTagBFP2 tag (blue). Green fluorescence with a 488 nm laser excitation, blue fluorescence with a 405 nm laser excitation, and phase-contrast imaging were recorded. Zeiss Zen were used to analyze the images.

### Reporting summary

Further information on research design is available in the Nature Portfolio Reporting Summary linked to this article.

## Data availability

The cryo-EM maps have been deposited in the Electron Microscopy Data Bank (EMDB) under accession codes EMD-41590 for portal, EMD-37155 for gp5-neck fiber, EMD-37154 for neck, EMD-37153 for sheath-tube, EMD-37151 for baseplate, EMD-37152 for tail fiber and EMD-37150 for gp33 CBD. The atomic coordinates have been deposited in the Protein Data Bank (PDB) under accession codes 8TS6 for portal, 8KEG for gp5-neck fiber, 8KEF for neck, 8KEE for sheath-tube, 8KEA for baseplate, 8KEC for tail fiber and 8KE9 for gp33 CBD. The genome of the cyanophage A-1(L) reported in this study have been deposited in the GenBank database with the accession number of OR360731. All other data supporting the findings in this study are included in the main article and associated files. The source data underlying Figs. 3b, d, f, g, 4d, 5c, d, and Supplementary Figs. 8d–h, 11a–d are provided as a Source Data file. Source data are provided with this paper.

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

## Acknowledgements

We thank Dr. Yong-Xiang Gao for technical support on cryo-EM data collection at the Cryo-EM Center at the University of Science and Technology of China (USTC). We also thank Dr. Zhen-Bang Liu and Gao Wu at the Core Facility Center for Life Sciences at USTC, for the technical assistance with confocal imaging and mass spectrometry, respectively. This work was supported by the National Natural Science Foundation of China (grant number U19A2020, C.-Z.Z.), the Ministry of Science and Technology of China (grant number 2018YFA0903100, C.-Z.Z.), and the Fundamental Research Funds for the Central Universities (grant number WK2070000195, Q.L.).

## Author contributions

C.-Z.Z. and Q.L. conceived, designed, and supervised the project. R.-C.Y., Q.L., C.-Z.Z. and Y.C. analyzed the data. R.-C.Y., Q.L. and C.-Z.Z. wrote and revised the manuscript. R.-C.Y., H.-Y.Z., N.C. and J.Z. performed A-1(L) purification and genome analysis. R.-C.Y., F.Y., P.H. and K.D. performed the cryo-EM data acquisition and data processing. X.X. provided the original A-1(L) virion. All of the authors discussed the data and read the manuscript.

## Competing interests

The authors declare no competing interests.
