## [Peer Review File · Nature Communications]

Structure of the intact tail machine of *Anabaena myophage* A-1(L)Reviewer #1 (Remarks to the Author):

The MS "Structure of Anabaena myophage A-1(L) reveals the assembly pattern of a multi-functional tail machine" describes the structure and particle composition of cyanophage A-1(L) and brief functional analysis of its host cell-binding proteins. High resolution cryo-EM analysis revealed errors in A-1(L) genome annotation and sequence, so the authors resequenced and reannotated it.

The main positive of the MS is that it is well illustrated when both the main text and supplementary figures are taken into account. The figures show important features at the level of domains and subdomains (instead of residues) as is appropriate for structures of this size. However, the figures and text is devoid of mentioning residue numbers. I strongly suggest (almost insist) that residue numbers flanking important domains are stated in the main text and indicated in the figures. For example, we do not know how many and which C-terminal residues of the TMP are presented in the figures.

The main drawback of the MS is that this expert in phage structure and function found little new information regarding phage tail organization or function. All features of the baseplate, tail tube, and tail sheath have been described earlier in other systems - R2 pyocin, T4 baseplate, PVCs/AFPs, or the type VI secretion system - many years ago. The structure of A-1(L) does contain truly interesting new information, but the authors do not cover it in any detail (more on this below).

Furthermore, the MS does not describe any work that examined the "assembly" or "assembly pattern" of the phage particle, contrary to the title and to the general flow of the text, which presents in the structure in some sort of an assembly sequence. The assembly has not been studied in this work because one structure of a complex, no matter the resolution, cannot reveal its "assembly" or "assembly pattern". An example of recent work that reveals some aspects of the assembly (but not the entire process!) of a tail-like structure is shown in Fig. 7 in PMID: 30905475 (Cell. 2019 Apr 4;177(2):370-383.e15. doi: 10.1016/j.cell.2019.02.020. Epub 2019 Mar 21). There is, of course, much more comprehensive work on the T4 tail assembly, which started 70 years ago and it is still incomplete.

In regards to the actual assembly, the authors possess one (small) system, where they can study assembly. That system is the neck fiber. The authors expressed and purified the neck fiber as a contiguous cluster of three genes. However, how the three proteins interact to form the fiber is unknown. Why there is a total of 14 copies of gp81 trimers per fiber - why not 6 or 26? Is there a tape measure protein that controls the length? Is gp80 required for the folding of gp81? Does gp81 serve as an assembly chaperone for gp82 i.e. required for trimerization? Or, perhaps, gp80 is that chaperone? These questions can be answered by expressing these proteins individually and in various pairs. Here is a recent example of a similar work PMID: 31325442 (J Mol Biol. 2019 Sep 6;431(19):3718-3739. doi: 10.1016/j.jmb.2019.07.022. Epub 2019 Jul 17). My point again - to claim "assembly" one must study ASSEMBLY, not just solve a structure.

Concerning the actual novelty of the work, there are several aspects concerning the structure of A-1(L) that this reviewer finds interesting and intriguing. Unfortunately, none of them is covered in sufficient detail to be fully understood. Instead (as already mentioned above) the paper repeats things that are essentially identical to the organization of other Contractile Injection Systems (sheath and tube organization, baseplate structure, etc.).

1. The 15-fold symmetry of gp5 is notable. The authors called this protein "Adaptor", but it is, in fact, a neck protein as is gp7 - they are Neck 1 and Neck 2 proteins (or vice versa - there is no established nomenclature on this). More on the nomenclature later. The paper does not explain how gp5 binds to the portal protein and how its 15-fold symmetry switches to 6-fold symmetry of gp7 (Neck 2 protein). Of note, the neck "decoration" or "Neck sheath" protein of phage Milano (Sonani, R.R., Esteves, N.C., Horton, A.A. et al. Neck and capsid architecture of the robust Agrobacterium phage Milano. Commun Biol 6, 921 (2023). <https://doi.org/10.1038/s42003-023-05292-1>) is 15-fold symmetric, even though

its Neck 1 protein, a gp5 equivalent, is 12-fold symmetric. It would be very interesting to compare the structure of gp5 with other Neck 1-like proteins.

2. The structure and copy number of the tape measure protein gp13. Apparently, the cryoEM map is good for building a C-terminal alpha-helix of gp15, but the paper does not state which fragment. The figures show 6 strands of density that can be interpreted as alpha-helices, although the map is not good enough for model building. So, it looks like there are 6 copies of TMP inside the tube channel. However, only 3 copies of the TMP can be seen interacting with the central spike. The missing copies of the TMP are not explained. Also, the structure and organization of the TMP needs to be compared to those of TMP of siphophages (a few have been published recently).

3. The organization of the TFs is very interesting. The garland of "beta-ring" domains of gp33 likely breaks when the sheath contracts. Perhaps, this is the first step in the conformational switch of the baseplate, which initiates sheath contraction. A low resolution reconstruction calculated using contracted tail particles that are usually present in phage samples would inform about the structure of the baseplate in the contracted state.

4. The second interesting point about the TFs is that host-cell binding experiments show that both fibers bind to the same host. Even though many T4-like phages employ different fibers for binding to the host cell surface, this is a rather uncommon property. SPO1-like phages (e.g. *Listeria* phage A511) also use multiple fibers/"tailspikes" when binding to the host cell surface, but, again, the mechanism is not universal. CBA120-like phages, which are fundamentally, T4-like, carry multiple tailspikes (essentially TF equivalents) but they use different tailspikes for binding to different host cells. A-1(L), similar to T4, appears to use two different TFs for host cell binding, which implies coordination of their action/deployment, similar to T4 TFs. To this end, however, I do not think that calling head-proximal and head-distal fibers of A-1(L) LTF and STF makes sense. LTF and STF stand for "long" and "short" tail fiber, the acronyms which weren't explained in the text. Regardless, the LTF gene of A-1(L) (gene 34) is actually shorter than the STF gene (gene 34). Unless the gp34 fiber is indeed much longer than the gp33 fiber (which I cannot tell from the figures), calling the former "LTF" does not make sense. It might be more appropriate to call the two fibers TF1 and TF2, especially considering the fact that we do not know which of those deploys first, unlike in T4. To this end, a competition binding experiment between gp33 and gp34 labeled differently would be of great interest.

5. The Discussion section mentions that the neck fiber did not bind to host cells (line 334-335). Nothing regarding this is presented in the Results section.

A few remarks concerning the nomenclature and general terminology used in the MS. A long tailed phage particle consists of a capsid (or head), neck, and tail. The baseplate is a part of the tail, not a separate component. Host cell receptor-binding proteins are often separated into the fourth category - they are either fibers or tailspikes and the latter are often considered to be a part of the tail baseplate (maybe wrongly so). The reason for this division is that these components are assembled independently from each other, except for the neck. The neck does not assemble on its own. Assembly of the neck requires that the head is packaged with DNA. Only a fully assembled tail binds to the neck.

Below are my line-by-line comments that aim to address points that require clarification.

The title

1-2. As mentioned above, the assembly was not studied in this work, so the words "assembly pattern" in the title are misleading.

The Abstract

34-35. "These findings elucidate the assembly pattern..." Again, no assembly was studied here.

The Intro.

66 and 78. "assembly pattern".

Results

98-99. 880-Å-long contractile tail, and a complicated baseplate at the distal end. See my remarks above. The baseplate is a tail component. The sentence, as written, is incorrect.

105. "the highly dynamic components gp80 and gp81". Not sure what this means. Maybe "highly mobile", but this does not make much sense either.

106. "a channel for the pass of genomic DNA," for the passage of DNA

113. "gp25 inserting". Probably, "inserted" is meant here, but this is incorrect as only a very small part of the spike is interacts with the hub. It is not "inserted" per se.

115. "further stabilized by six plugs/monomeric gp16 (Fig. 1b)". I am not sure why this LysM domain-containing protein is called a "plug". It does not plug anything.

120. "The neck of A-1(L) is assembled on a pentadecameric adaptor". This is not true. 1. We do not know what is assembled onto what. 2. The "pentadecameric adaptor" is, in fact, a neck protein.

126. "two ends of the dsDNA genome in our model". Which two ends? The ring of density around the portal is most likely DNA, but I do not understand why the authors think it is one of the genome's termini. Secondly, where the DNA ends and where the TMP starts in the tail tube is unclear, and the experimental data that supports the proposed segmentation is not provided.

138 "accommodates one end of the genomic dsDNA (Supplementary Fig. 1b)." I do not see any "ends of DNA" anywhere there.

147 "The neck is assembled on the adaptor," - Nonsense.

149 "Loopα4-α5 of 15" - Do not know what alpha4-alpha5 is. Residue numbers would be useful here.

195 "The assembly of ~880-Å-long tail of A-1(L) is initiated by the tube". Here we go about the assembly. We do not know anything about the assembly of this phage, and this paper does not present any data about the assembly.

202 "Encircling the TMP, the tube forms a six-entry helical structure with" - Six-start helical structure, perhaps? I have never heard about a six-entry helical structure... Or, maybe sixfold symmetric helical structure?

208. Six-entry again.

208 Different from the previously reported phages. - Not true. See here, for example:
https://link.springer.com/chapter/10.1007/978-1-4614-0980-9_5 [SEP]

222 "initiating the growth of sheath"... 1. The sheath is not a plant - it can't "grow". 2. No data presented here supports this claim.

233 "baseplate (Fig. 5a). It provides a platform for the growth of the tail and the attachment of tail fibers." Wrong on many levels. The baseplate is a component of the tail, it cannot provide a platform for the "growth" of it. The tail is not a plant, as mentioned above. And, for a 1000th time, there it no data to support this claim.

237 "The N-terminal helices form an inverted tripod, which interwinds with the vertical tripod of three TMP C-terminal helices (Supplementary Fig. 5b)." I do not understand the distinction between "inverted" and "vertical" tripods...

240 "Supplementary Fig. 5c), each subunit of which adopts an architecture different from previously reported hub proteins (Supplementary Fig. 5d)." Not true. T4 gp6 has two insertion domains.

246-251. Activity assays for the hub domains. The method used can apparently detect an increase in cytochrome-caused solution absorption at a level below 1 mAu. A reference describing this technique is required. Or, if this assay is new, a positive control is required, for example, cell lysis caused by a lysozyme in the same conditions.

270 "A-1(L) possesses two types of tail fibers: six LTFs/gp34 trimers and six STFs/gp33". First of all, LTF and STF must be spelled out. Secondly, as noted above, the terms LTF and STF are questionable when applied to A-1(L).

274 "The shoulder domain of LTF is anchored to the wedge of baseplate core," - I am not sure what "baseplate core" is in this phage. T4 baseplate can be divided into a universally "core" and lots of additional non-universally conserved domains. In this phage, the entire baseplate is equivalent to the universally conserved core of the T4 baseplate.

280. "baseplate core" again.

294 "a couple of" cannot be used in a formal document such as a scientific article.

Discussion

319 "Myoviridae cyanophage A-1(L), and clearly elucidated its fine assembly pattern." - Nope, this is not true. This scientist sees no "revelations" of any "fine assembly patterns".

319-322 "After genomic DNA packaging, the pentadecameric adaptor and hexameric connector are sequentially recruited to the dodecameric portal that is perfectly accommodated to a 5-fold vertex of icosahedral capsid, forming a symmetry-mismatched neck." - Once again, a misleading statement. Nothing of this sort has been studied in this MS.

334 "Notably, the distal CBM module in gp80 of the neck fiber does not possess binding activity toward *Anabaena* sp. PCC 7120," - Spell out CBM. This isn't shown anywhere in the Results.

338 "The phage T4 was proposed to harbor a complicated baseplate" - What does "proposed" mean here? Isn't the structure of the T4 baseplate solved and its atomic model is available? Is not complicated enough?

350-351 "For myophages, the β -helix of central spike might act as a drill to penetrate the cell outer membrane." - Wrong reference. These two are more appropriate: PMID: 11823865 and PMID: 22325780

Figures.

Once again, residue numbers are required in strategic positions (i.e. boundaries of important domains discussed in the text).

Fig. 5 and 6 panels are too small to be understood. And this is for a person who had studied all the available homologous structures.

In all figures and the text - a phage particle does not have a top or bottom. It is not a building. The two ends of the tail can be referred to, for example, head-proximal and head-distal.

Supplementary Fig. 2. A color scale bar of e-potential must be shown in kT/e. It must be explicitly stated whether the same color scale bar range is used in all panels.

Supplementary Fig. 8a. A larger fragment of the micrograph needs to be shown to convince the viewer that the round electron-dense feature in the lower part of the image is a lipid vesicle but not a carbon film.

Reviewer #2 (Remarks to the Author):

Yu, R. et al. determined the cryoEM structure of the contractile tail in myophage A-1(L) and described the details on the architecture of the A-1(L) tail. With structural information, they compared some tail proteins with homologs in other phages, identified unique domains in the hub, and explored the functions via *in vitro* enzymatic assays. They also characterized two types of tail fibers involved in the viral attachment on the host surface. In addition, they re-annotated the phage genome based on the structural information. The work presented in paper was extensive and quite encouraging. However, the writing and the structural presentation style are somewhat confused, thus it still requires lots of efforts to polish before publication. I therefore agree to publish paper, however, after proper (and complete) revisions.

Here is list of my concerns that should be addressed, which is believed to improve the quality of the paper.

Major comments:

Authors should put lots of efforts on manuscript polishing, including text and figures. Some vocabularies (e.g., "configuration", "assembly pattern", "fine") are confused and not proper. Some descriptions are not showed in details on the corresponding figures. For figures, authors used segmented density maps to present the overall structure of each parts, which is confused and hard to see the details. I highly recommend authors to re-prepare all figures and show the atomic model instead of density map.

Authors have detailed descriptions on individual tail elements in A-1(L) myophage. However, I felt that authors loss connections with other contractile tailed phages and bacterial contractile injection systems. The structural comparison would really benefit to the field, especially for re-engineering. I therefore encourage authors to make a detailed comparison to illustrate the conserved/diverse elements among different apparatus.

Minor comments:

Title:

"assembly pattern" is really misleading and I was thinking that the paper is mainly about the tail assembly. I recommended to use other word, like architecture. In addition, I did not see that the tail has multiple function from text, which is also mentioned in discussion part.

Introduction part:

Line 47-49: "but generally consist of an icosahedral capsid encapsulating a dsDNA genome, followed by a complicated tail machine." Please rewrite this sentence because the description here is mainly about the tailed phage and the reference cited here is about tailed bacteriophage, not general phages. In addition, some capsids of tailed phage are prolate, not icosahedral.

Line 53-55: the sentences are confused. Please rewrite to make them easier to follow.

Line 57: "refer to" change to "assemble into"?

Line 62: "Dunne et. al" change to "Dunne et al. "

Line 77: "fine" change to "high resolution"?

Line 83-84: "both two types of tail fibers are folded back and pack against the sheath." If I understood correctly from the result part, only one type of tail fiber fold back and interacts with sheath. Then the overall structure is similar to T4 (See Hu, B., et al. 2015. PNAS). Please address it.

Result part:

Supplementary figure 9-11 and supplementary table 3 are not referred in paper.

Line 91-92: "the results of mass spectrometry showed that there exist 20 extra structural proteins." Please provide the corresponding data. In addition, table S2 looks like to contain some mass spectrometry data, but authors did not cite it.

Line 94: "successive operons" It's difficult to understand. Please change with proper word.

Line 96: "indicating that they are independently regulated." Do you have any data to show that the expressions of gp80-gp82 are independently regulated? Please address it.

Line 103-104: "around which five neck fibers are attached (Fig. 1b)." It's not clear to see five neck fibers attaching on the neck part based on side view. Please show the top/bottom view.

Line 105-106: "followed by the highly dynamic components gp80 and gp81 (Fig. 1b)." The corresponding contents are not shown in Fig. 1b.

Line 116-117: "six short tail fibers (STFs, gp33 trimers) are folded back and pack against the sheath." I do not see clear contacts between short tail fibers and sheath. Please show the details.

Line 117-118: "six plugs, which adopt two configurations," please reword "configurations" and I can not see two conformations of plugs from Fig. 1b. Please show the details if authors want to highlight in the text.

The organization of the description part on neck is really confused and please re-write this part properly. I recommend to describe the structures following the order from top to bottom (portal, neck, and then adaptor).

Line 124-126: authors observed "two extra densities" and assigned them to "two ends of the dsDNA genome". Please highlight two extra densities in Supplementary Fig. 1 a and do authors use "two ends of the dsDNA" to means 5'- and 3'- end of dsDNA? If that case, how do authors know without determining the overall structure of viral genome? Please address it.

Line 132: "polymerize" generally means that the proteins assemble into super-structure, like capsid or filament. Please reword it.

Line 139-140: "Considering that all structure-known portal-adaptor complex are 12-fold symmetric" please cite the corresponding papers.

Line 141: "but failed in obtaining a best three-dimensional classification." Change "best" to "reasonable" and did author try 3D classification here? If not, why use "classification", not "reconstruction"?

Line 144-146: authors mentioned that there is not fixed orientation between portal and adaptor. Could authors validate with detailed analysis? For example, authors could calculate minimal absolute

different in azimuth angle based on the orientations determined from portal and adaptor.

Line 147-148: "The neck is assembled on the adaptor, which coordinates the interactions with the portal, the connector and neck fibers." The sentence is confused, because authors mentioned that portal, adaptor, and connector assemble to the neck (Line 102-103). Then how neck is assembled on the adaptor?

Line 150: "whereas three adaptor subunits tightly interact with the trimeric gp82N via the interface between Loop ^{α 1- α 2} and Loop ^{α 5- α 6}." How "tightly" interactions? Are there some salt bridges? Please show the details on Fig. 2. In addition, please mention clearly the Loop ^{β 1- β 2} of which protein interacts with the Loop ^{β 5- β 6} of which protein.

Line 165-166: "gp80 and gp81, which are encoded in a same operon with gp82, constitute the neck fiber." If I understand correctly, the neck fiber includes gp80, gp81, and gp82, not only gp80 and gp81. The sentence is confused and please reword it.

Line 169: "As shown in the electrophoresis gel" Authors here showed the gel of the purified gp82-gp81-gp80 complex, but did not show the gel of the purified phage to confirm the stoichiometric ratios.

Line 176: "As expected, gp82 indeed possesses two separate domains gp82N and gp82C" What did authors expect? Please clarify or rewrite the sentence.

Line 181-183: "the modelled structure of gp80 trimer of relatively larger size should be localized to the most distal of the neck fiber." How did author make the conclusion of the location of protein domain only based on the size? Please rewrite the sentence.

Line 199-200: I am curious whether the structure of TMP follows the helical arrangement of tube and whether they have some contacts or not.

Line 201: please show the detailed interactions between TMP and baseplate in Supplementary Fig. 4a.

Line 205-206: I can not see the β -hairpin of inner tube protruding towards the consecutive inner tube ring in Supplementary Fig. 4b. Please re-prepare the figure.

Line 226-227: It should be "disrupts" and "terminates".

Line 244: please change to "name the three domains as lysozyme, endoglucanase, and peptidase domain, respectively".

Line 248: "either yield much lower expression level of the recombinant protein" Fig. 5c-d do not show the corresponding data. Please provide the details.

Line 262-263: it's not clear to see the interactions between the N-terminal helical bundle of wedge and the LysM domain of the plug. Please re-prepare the proper figure to show the details.

Line 280-283: authors described the structure of STF. However, I did not see any descriptions about the contacts between STF and sheath (also from figure). Therefore, I am not sure that STF attaches on the sheath as authors stated in text. Please address it and tune down the statements about the unique features of tail fibers in A-1(L) compared to other phages.

Line 289-291: authors performed *in vitro* binding assays using the purified full-length LTF and STF, and showed that both proteins could bind to the cell surface. However, I suggest that authors should use the arm domain of LTF and the CBD domain of STF instead of full-length proteins.

Discussion part:

Line 334-335: please show the data or cite the reference about the statement "the distal CBM module in gp80 of the neck fiber does not possess binding activity...".

Line 348: change "versicles" to "vesicles".

Line 352-353: please cite the PDB entry and the corresponding reference when using the structure for analysis.

Line 360: I did not see multiple functions of the tail from the manuscript.

Line 363-364: authors state that A-1(L) could be an ideal cyanophage for the applications in synthetic biology. Is it available for genetic manipulation or re-engineering in A-1(L) phage? Since some phage systems could be re-engineered to target different hosts, what's the advantage of A-1(L) compared to the established other phages?

Figure 1:

- a. I recommend to combine the two DNA organization schemes into one big panel.
- b. I highly recommend to re-prepare the figure with proper color annotation. I do not see any points why two different colors are applied to sheath and tube. Moreover, some positions of structural proteins mentioned in text are not clear (e.g., tube initiator, sheath initiator), please highlight by changing colors or showing the details with zoom-in figures.

Figure 2:

d-f. The labels are overlapped with figure. Please fix it. Moreover, it's hard to distinguish different protomers from figure (e.g., three gp82 proteins bind to adaptor, but they look like one protein from figure). Please re-prepare figures with different colors on each protomers.

Figure 3:

- a. It's difficult to see five neck fibers protruding from one phage. Please highlight it.
- d. it's difficult to understand how authors deduce the molecular weight from the plot. It would be better to indicate the predicted molecular weight from the peak via dashed line.

Figure 4:

I highly recommend authors to re-prepare the figure and show the overall structures using atomic model instead of segmented map. In addition, I still did not see points why authors use two different colors for sheath and tube. For panel a, I cannot see clearly the sheath S2 N-tail because the color is too similar to tube.

Figure 5:

f-h. I cannot see the interaction details. Please highlight by showing zoom-in structures, not just remarked with red circles.

Figure 6:

e. It's hard to distinguish different domains of STF. Please color the STF based on the domains shown in panel c.

Supplementary Figure 2:

authors stated that the electrostatic potentials on each neck elements are complementary. However, I cannot see the interface boundary between each neck components. Please highlight them in the figure.

Supplementary Figure 3:

please re-organize the panels. For panel c, I would like to show the diameter difference between gp80 and gp81, instead of showing structural superposition.

Supplementary Figure 4:

c. please cite the corresponding PDB entries.

Supplementary Figure 5:

e: all domain annotations miss "domain".

We appreciated very much the two reviewers' constructive comments and suggestions, which help us in depth to improve the quality of our manuscript. We have addressed all questions and suggestions point by point.

Reviewer #1

The MS "Structure of *Anabaena* myophage A-1(L) reveals the assembly pattern of a multi-functional tail machine" describes the structure and particle composition of cyanophage A-1(L) and brief functional analysis of its host cell-binding proteins. High resolution cryo-EM analysis revealed errors in A-1(L) genome annotation and sequence, so the authors resequenced and reannotated it.

The main positive of the MS is that it is well illustrated when both the main text and supplementary figures are taken into account. The figures show important features at the level of domains and subdomains (instead of residues) as is appropriate for structures of this size. However, the figures and text are devoid of mentioning residue numbers. I strongly suggest (almost insist) that residue numbers flanking important domains are stated in the main text and indicated in the figures. For example, we do not know how many and which C-terminal residues of the TMP are presented in the figures.

The main drawback of the MS is that this expert in phage structure and function found little new information regarding phage tail organization or function. All features of the baseplate, tail tube, and tail sheath have been described earlier in other systems - R2 pyocin, T4 baseplate, PVCs/AFP, or the type VI secretion system - many years ago. The structure of A-1(L) does contain truly interesting new information, but the authors

do not cover it in any detail (more on this below).

Furthermore, the MS does not describe any work that examined the “assembly” or “assembly pattern” of the phage particle, contrary to the title and to the general flow of the text, which presents in the structure in some sort of an assembly sequence. The assembly has not been studied in this work because one structure of a complex, no matter the resolution, cannot reveal its “assembly” or “assembly pattern”. An example of recent work that reveals some aspects of the assembly (but not the entire process!) of a tail-like structure is shown in Fig. 7 in PMID: 30905475 (Cell. 2019 Apr 4;177(2):370-383.e15. doi: 10.1016/j.cell.2019.02.020. Epub 2019 Mar 21). There is, of course, much more comprehensive work on the T4 tail assembly, which started 70 years ago and it is still incomplete.

In regards to the actual assembly, the authors possess one (small) system, where they can study assembly. That system is the neck fiber. The authors expressed and purified the neck fiber as a contiguous cluster of three genes. However, how the three proteins interact to form the fiber is unknown. Why there is a total of 14 copies of gp81 trimers per fiber - why not 6 or 26? Is there a tape measure protein that controls the length? Is gp80 required for the folding of gp81? Does gp81 serve as an assembly chaperone for gp82 i.e. required for trimerization? Or, perhaps, gp80 is that chaperone? These questions can be answered by expressing these proteins individually and in various pairs. Here is a recent example of a similar work PMID: 31325442 (J Mol Biol. 2019 Sep 6;431(19):3718-3739. doi: 10.1016/j.jmb.2019.07.022. Epub 2019 Jul 17).

My point again - to claim “assembly” one must study ASSEMBLY, not just solve a

structure.

A: Thanks for your suggestion. We have revised the description concerning the assembly pattern throughout the manuscript.

A total of 14 copies of gp81 trimers per fiber were deduced from the molecular weight of the recombinant neck fiber (Fig. 3d), which is further supported by the statistical analysis against the repetitive bead number in dozens of in situ ~100-nm-long fibers. According to your suggestion, we performed a preliminary study on the assembly of the neck fiber. The recombinant gp82-gp81 and gp81-gp80 were purified and applied to the negative-staining EM, respectively. The results showed that both complexes of gp82-gp81 and gp81-gp80 could form a bead-chain-like structure as well (Fig. 3f-g), suggesting that the neck fiber is independently self-assembled in prior of being hooked to the neck. Further studies are needed to elucidate the detailed assembly process of the neck fiber.

Concerning the actual novelty of the work, there are several aspects concerning the structure of A-1(L) that this reviewer finds interesting and intriguing. Unfortunately, none of them is covered in sufficient detail to be fully understood. Instead (as already mentioned above) the paper repeats things that are essentially identical to the organization of other Contractile Injection Systems (sheath and tube organization, baseplate structure, etc.).

Q1. The 15-fold symmetry of gp5 is notable. The authors called this protein “Adaptor”, but it is, in fact, a neck protein as is gp7 - they are Neck 1 and Neck 2 proteins (or vice

versa - there is no established nomenclature on this). More on the nomenclature later. The paper does not explain how gp5 binds to the portal protein and how its 15-fold symmetry switches to 6-fold symmetry of gp7 (Neck 2 protein). Of note, the neck “decoration” or “Neck sheath” protein of phage Milano (Sonani, R.R., Esteves, N.C., Horton, A.A. et al. Neck and capsid architecture of the robust Agrobacterium phage Milano. *Commun Biol* 6, 921 (2023). <https://doi.org/10.1038/s42003-023-05292-1>) is 15-fold symmetric, even though its Neck 1 protein, a gp5 equivalent, is 12-fold symmetric. It would be very interesting to compare the structure of gp5 with other Neck 1-like proteins.

A: Yes, gp5 and gp7 are both components of the A-1(L) neck. We tried to determine the interface between the neck/gp5 and portal/gp2 in our cryo-EM map via rotating the structures and calculating the clashscores, but failed in obtaining a pose with the lowest clashscore, which represents the most favored relative orientation. It suggested that the difference between the relative rotations of 30° and 24° (for the 12-fold and 15-fold symmetric gp2 and gp5, respectively) is not distinguishable at the present resolution; however, the two components are docked to each other via an interface which is generally complementary in electrostatic potential and shape.

As shown in Fig. 2d, the 15-fold symmetry of neck/gp5 switches to 6-fold symmetry of neck/gp7 mainly via the three-repeated interactions between Loop α 4- α 5 of five gp5 subunits and α 1 helices of two gp7 subunits.

We have added the structural comparison of neck/gp5 against those in the previously reported phages as the Supplementary Fig. 1b. Despite the PDB entry of the

neck 1 structure of *Agrobacterium* phage Milano has not yet been released, a unique 15-fold symmetric collar sheath protein in the neck of the phage Milano has been cited in our revised manuscript.

Q2. The structure and copy number of the tape measure protein gp13. Apparently, the cryo-EM map is good for building a C-terminal alpha-helix of gp13, but the paper does not state which fragment. The figures show 6 strands of density that can be interpreted as alpha-helices, although the map is not good enough for model building. So, it looks like there are 6 copies of TMP inside the tube channel. However, only 3 copies of the TMP can be seen interacting with the central spike. The missing copies of the TMP are not explained. Also, the structure and organization of the TMP needs to be compared to those of TMP of siphophages (a few have been published recently).

A: The modeled C-terminal helix corresponds to the residues Pro666~Ala689 of TMP/gp13. We have added the residue number of this C-terminal helix in the revised manuscript according to your suggestion. Moreover, we have discussed the structure and organization of the TMP and the missing C-terminal copies of TMP in the “Discussion” section of the revised manuscript.

Q3. The organization of the TFs is very interesting. The garland of “beta-ring” domains of gp33 likely breaks when the sheath contracts. Perhaps, this is the first step in the conformational switch of the baseplate, which initiates sheath contraction. A low-resolution reconstruction calculated using contracted tail particles that are usually

present in phage samples would inform about the structure of the baseplate in the contracted state.

A: Thanks for your suggestion. Yes, the contracted structure of A-1(L) could elucidate the conformational switch of the baseplate and the tail fibers. However, ~2700 contracted particles in our cryo-EM data are not enough for calculating the structure of A-1(L) in the contracted conformation at a reasonable resolution. We will put more efforts to produce enough contracted A-1(L) particles for structure determination.

Q4. The second interesting point about the TFs is that host-cell binding experiments show that both fibers bind to the same host. Even though many T4-like phages employ different fibers for binding to the host cell surface, this is a rather uncommon property. SPO1-like phages (e.g. *Listeria* phage A511) also use multiple fibers/“tailspikes” when binding to the host cell surface, but, again, the mechanism is not universal. CBA120-like phages, which are fundamentally, T4-like, carry multiple tailspikes (essentially TF equivalents) but they use different tailspikes for binding to different host cells. A-1(L), similar to T4, appears to use two different TFs for host cell binding, which implies coordination of their action/deployment, similar to T4 TFs. To this end, however, I do not think that calling head-proximal and head-distal fibers of A-1(L) LTF and STF makes sense. LTF and STF stand for “long” and “short” tail fiber, the acronyms which weren't explained in the text. Regardless, the LTF gene of A-1(L) (gene 34) is actually shorter than the STF gene (gene 34). Unless the gp34 fiber is indeed much longer than the gp33 fiber (which I cannot tell from the figures), calling the former “LTF” does not

make sense. It might be more appropriate to call the two fibers TF1 and TF2, especially considering the fact that we do not know which of those deploys first, unlike in T4. To this end, a competition binding experiment between gp33 and gp34 labeled differently would be of great interest.

A: The full names of LTF and STF were indicated in the last paragraph of “Overall structure of A-1(L) tripartite tail machine”. Structural analysis showed that gp34 trimer possesses a length of 274 Å, whereas gp33 trimer is of 258 Å. Accordingly, gp34 and gp33 are termed as the LTF and STF, respectively. We have indicated the length of both fibers in the revised text and the Supplementary Fig. 10a.

According to your suggestion, we performed competition binding assays between gp33 and gp34, which were fused with eGFP (green) and pmTagBFP2 (blue), respectively. The results showed that both fibers could simultaneously bind to the surface of host cells (Fig. 6d and Supplementary Fig. 10c). It suggested that the two fibers might recognize and bind to receptors on the host cell surface. Further investigations on the infection of A-1(L) against its host *Anabaena* sp. PCC 7120 would help us to reveal which kind of fiber deploys first upon infection.

Q5. The Discussion section mentions that the neck fiber did not bind to host cells (line 334-335). Nothing regarding this is presented in the Results section.

A: Added as the Supplementary Fig. 10d.

A few remarks concerning the nomenclature and general terminology used in the

MS. A long-tailed phage particle consists of a capsid (or head), neck, and tail. The baseplate is a part of the tail, not a separate component. Host cell receptor-binding proteins are often separated into the fourth category - they are either fibers or tailspikes and the latter are often considered to be a part of the tail baseplate (maybe wrongly so). The reason for this division is that these components are assembled independently from each other, except for the neck. The neck does not assemble on its own. Assembly of the neck requires that the head is packaged with DNA. Only a fully assembled tail binds to the neck.

A: We have reorganized our structural description according to your suggestions.

Below are my line-by-line comments that aim to address points that require clarification.

The title

1-2. As mentioned above, the assembly was not studied in this work, so the words “assembly pattern” in the title are misleading.

A: Revised.

The Abstract

34-35. “These findings elucidate the assembly pattern...” Again, no assembly was studied here.

A: Revised.

The Intro.

66 and 78. “assembly pattern”.

A: Revised.

Results

98-99. 880-Å-long contractile tail, and a complicated baseplate at the distal end. See my remarks above. The baseplate is a tail component. The sentence, as written, is incorrect.

105. “the highly dynamic components gp80 and gp81”. Not sure what this means. Maybe “highly mobile”, but this does not make much sense either.

106. “a channel for the pass of genomic DNA,” for the passage of DNA.

113. “gp25 inserting”. Probably, “inserted” is meant here, but this is incorrect as only a very small part of the spike interacts with the hub. It is not “inserted” per se.

A: All above points have been revised.

115. “further stabilized by six plugs/monomeric gp16 (Fig. 1b).”. I am not sure why this LysM domain-containing protein is called a “plug”. It does not plug anything.

A: We simply call it gp16 in the revised manuscript.

120. “The neck of A-1(L) is assembled on a pentadecameric adaptor”. This is not true.

1. We do not know what is assembled onto what. 2. The “pentadecameric adaptor” is, in fact, a neck protein.

126. “two ends of the dsDNA genome in our model”. Which two ends? The ring of density around the portal is most likely DNA, but I do not understand why the authors think it is one of the genome’s termini. Secondly, where the DNA ends and where the TMP starts in the tail tube is unclear, and the experimental data that supports the proposed segmentation is not provided.

138. “accommodates one end of the genomic dsDNA (Supplementary Fig. 1b).” I do not see any “ends of DNA” anywhere there.

147 “The neck is assembled on the adaptor,” - Nonsense.

149. “Loop α 4- α 5 of 15” - Do not know what alpha4-alpha5 is. Residue numbers would be useful here.

195. “The assembly of ~880-Å-long tail of A-1(L) is initiated by the tube”. Here we go about the assembly. We do not know anything about the assembly of this phage, and this paper does not present any data about the assembly.

202. “Encircling the TMP, the tube forms a six-entry helical structure with” - Six-start helical structure, perhaps? I have never heard about a six-entry helical structure... Or, maybe sixfold symmetric helical structure?

208. Six-entry again.

208. Different from the previously reported phages. - Not true. See here, for example:
https://link.springer.com/chapter/10.1007/978-1-4614-0980-9_5

222. “initiating the growth of sheath”... 1. The sheath is not a plant - it can’t “grow”. 2. No data presented here supports this claim.

233. “baseplate (Fig. 5a). It provides a platform for the growth of the tail and the

attachment of tail fibers.” Wrong on many levels. The baseplate is a component of the tail, it cannot provide a platform for the “growth” of it. The tail is not a plant, as mentioned above. And, for a 1000th time, there is no data to support this claim.

237. “The N-terminal helices form an inverted tripod, which interwinds with the vertical tripod of three TMP C-terminal helices (Supplementary Fig. 5b).” I do not understand the distinction between “inverted” and “vertical” tripods...

A: All above points have been revised.

240. “Supplementary Fig. 5c), each subunit of which adopts an architecture different from previously reported hub proteins (Supplementary Fig. 5d).” Not true. T4 gp6 has two insertion domains.

A: As reported (Taylor, N.M., et al., Structure of the T4 baseplate and its function in triggering sheath contraction. Nature, 2016. 533(7603): p. 346-52.), T4 gp6 constitutes the baseplate wedge, whereas T4 gp27 is the hub. Superposition showed that T4 gp27 is structurally similar to the barrel domain of A-1(L) hub/gp15, with a root-mean-square-deviation of 3.1 Å over 276 C α atoms. It suggested that the barrel domain is conserved in phage hub proteins; moreover, A-1(L) gp15 possesses three extra domains.

Fig. R1 Superposition of T4 gp27 against A-1(L) gp15. The domains of gp15 are labeled and colored differently.

246-251. Activity assays for the hub domains. The method used can apparently detect an increase in cytochrome-caused solution absorption at a level below 1 mAu. A reference describing this technique is required. Or, if this assay is new, a positive control is required, for example, cell lysis caused by a lysozyme in the same conditions.

A: Sorry for the mistake. The Y axis stands for the Absorbance Unit (AU) at 610 nm of spectrophotometer. We have revised the corresponding figures, and applied the commercially available lysozyme as a positive control (Supplementary Fig. 8d).

270. "A-1(L) possesses two types of tail fibers: six LTFs/gp34 trimers and six

STFs/gp33". First of all, LTF and STF must be spelled out. Secondly, as noted above, the terms LTF and STF are questionable when applied to A-1(L).

A: The full names of LTF and STF were indicated in the last paragraph of "Overall structure of A-1(L) tripartite tail machine". For A-1(L), structural analysis showed that gp34 and gp33 trimers display a length of 274 and 258 Å, respectively. Accordingly, gp34 and gp33 are termed as the LTF and STF, respectively. We have labeled the length of both fibers in the text and the Supplementary Fig. 10a.

274. "The shoulder domain of LTF is anchored to the wedge of baseplate core," - I am not sure what "baseplate core" is in this phage. T4 baseplate can be divided into a universally "core" and lots of additional non-universally conserved domains. In this phage, the entire baseplate is equivalent to the universally conserved core of the T4 baseplate.

280. "baseplate core" again.

294. "a couple of" cannot be used in a formal document such as a scientific article.

A: All above points have been revised.

Discussion

319. "Myoviridae cyanophage A-1(L), and clearly elucidated its fine assembly pattern."

- Nope, this is not true. This scientist sees no "revelations" of any "fine assembly patterns".

319-322. "After genomic DNA packaging, the pentadecameric adaptor and hexameric

connector are sequentially recruited to the dodecameric portal that is perfectly accommodated to a 5-fold vertex of icosahedral capsid, forming a symmetry-mismatched neck.” - Once again, a misleading statement. Nothing of this sort has been studied in this MS.

334. “Notably, the distal CBM module in gp80 of the neck fiber does not possess binding activity toward *Anabaena* sp. PCC 7120,” - Spell out CBM. This isn’t shown anywhere in the Results.

338. “The phage T4 was proposed to harbor a complicated baseplate” - What does “proposed” mean here? Isn’t the structure of the T4 baseplate solved and its atomic model is available? Is not complicated enough?

350-351. “For myophages, the β -helix of central spike might act as a drill to penetrate the cell outer membrane²⁶. “- Wrong reference. These two are more appropriate: PMID: 11823865 and PMID: 22325780

A: All above points have been revised.

Figures.

Once again, residue numbers are required in strategic positions (i.e. boundaries of important domains discussed in the text).

A: Following your suggestion, we added residue numbers flanking important domains throughout the revised manuscript and figure legends.

Fig. 5 and 6 panels are too small to be understood. And this is for a person who had

studied all the available homologous structures.

In all figures and the text - a phage particle does not have a top or bottom. It is not a building. The two ends of the tail can be referred to, for example, head-proximal and head-distal.

Supplementary Fig. 2. A color scale bar of e-potential must be shown in kT/e. It must be explicitly stated whether the same color scale bar range is used in all panels.

Supplementary Fig. 8a. A larger fragment of the micrograph needs to be shown to convince the viewer that the round electron-dense feature in the lower part of the image is a lipid vesicle but not a carbon film.

A: All above points have been revised.

Reviewer #2

Yu, R. et al. determined the cryo-EM structure of the contractile tail in myophage A-1(L) and described the details on the architecture of the A-1(L) tail. With structural information, they compared some tail proteins with homologs in other phages, identified unique domains in the hub, and explored the functions via in vitro enzymatic assays. They also characterized two types of tail fibers involved in the viral attachment on the host surface. In addition, they re-annotated the phage genome based on the structural information. The work presented in paper was extensive and quite encouraging. However, the writing and the structural presentation style are somewhat confused, thus it still requires lots of efforts to polish before publication. I therefore agree to publish paper, however, after proper (and complete) revisions.

Here is list of my concerns that should be addressed, which is believed to improve the quality of the paper.

Major comments:

Q1. Authors should put lots of efforts on manuscript polishing, including text and figures. Some vocabularies (e.g., “configuration”, “assembly pattern”, “fine”) are confused and not proper. Some descriptions are not showed in details on the corresponding figures. For figures, authors used segmented density maps to present the overall structure of each part, which is confused and hard to see the details. I highly recommend authors to re-prepare all figures and show the atomic model instead of density map.

A: Thanks for your suggestions. We have carefully proofread the manuscript and corrected mistakes throughout the text. Also, we have revised all figures and provided more figures/panels to show the details according to your suggestion.

Q2. Authors have detailed descriptions on individual tail elements in A-1(L) myophage. However, I felt that authors loss connections with other contractile tailed phages and bacterial contractile injection systems. The structural comparison would really benefit to the field, especially for re-engineering. I therefore encourage authors to make a detailed comparison to illustrate the conserved/diverse elements among different apparatus.

A: According to your suggestion, we have added comparison of A-1(L) structural

components against those in the previously reported tailed phages and bacterial contractile injection systems (Supplementary Figs. 1, 6, 7 and 10a). The related description was also added in the “Result” section.

Minor comments:

Title:

“assembly pattern” is really misleading and I was thinking that the paper is mainly about the tail assembly. I recommended to use other word, like architecture. In addition, I did not see that the tail has multiple function from text, which is also mentioned in discussion part.

A: Revised.

Introduction part:

Line 47-49: “but generally consist of an icosahedral capsid encapsulating a dsDNA genome, followed by a complicated tail machine.” Please rewrite this sentence because the description here is mainly about the tailed phage and the reference cited here is about tailed bacteriophage, not general phages. In addition, some capsids of tailed phage are prolate, not icosahedral.

Line 53-55: the sentences are confused. Please rewrite to make them easier to follow.

Thus it limits the broad applications, which need to isolate phages against the highly polymorphic host bacteria and produce commercialized phage agents in large scale.

Line 57: “refer to” change to “assemble into”?

Line 62: “Dunne et. al” change to “Dunne et al.”

Line 77: "fine" change to "high resolution"?

A: All above points have been revised.

Line 83-84: “both two types of tail fibers are folded back and pack against the sheath.”

If I understood correctly from the result part, only one type of tail fiber fold back and interacts with sheath. Then the overall structure is similar to T4 (See Hu, B., et al. 2015. PNAS). Please address it.

A: Sorry for the unclear description. For T4, most LTFs are folded back against the tail sheath, whereas the six STFs are retracted at the baseplate plane. In our case, both types of tail fibers of A-1(L) are folded back, despite that only the LTFs directly interact with the sheath. We have revised the sentence, and tuned down the statements about the unique features of tail fibers in A-1(L) compared to other phages.

Result part:

Supplementary figure 9-11 and supplementary table 3 are not referred in paper.

A: Sorry for the mistake. We have cited the Supplementary figures 9-11 (now Supplementary Figs. 13-15 in the revised manuscript) and Supplementary Table 3 in the “Methods” section.

Line 91-92: “the results of mass spectrometry showed that there exist 20 extra structural proteins.” Please provide the corresponding data. In addition, table S2 looks like to

contain some mass spectrometry data, but authors did not cite it.

A: Thanks for your reminding. We have cited the Supplementary Table 2 (now Supplementary Table 1 in the revised manuscript).

Line 94: “successive operons” It’s difficult to understand. Please change with proper word.

Line 96: “indicating that they are independently regulated.” Do you have any data to show that the expressions of gp80-gp82 are independently regulated? Please address it.

A: These two points have been revised.

Line 103-104: “around which five neck fibers are attached (Fig. 1b).” It’s not clear to see five neck fibers attaching on the neck part based on side view. Please show the top/bottom view.

Line 105-106: “followed by the highly dynamic components gp80 and gp81 (Fig. 1b).” The corresponding contents are not shown in Fig. 1b.

A: Revised. The low-resolution densities of the remaining components of the flexible neck fiber were added to the Fig. 1b. Now, the five neck fibers could be seen on side view. In addition, a top view of five neck fibers were displayed in Fig. 2c.

Line 116-117: “six short tail fibers (STFs, gp33 trimers) are folded back and pack against the sheath.” I do not see clear contacts between short tail fibers and sheath.

Please show the details.

A: Sorry for the unclear description. Yes, the STFs do not directly interact with the sheath. We have revised the sentence.

Line 117-118: “six plugs, which adopt two configurations,” please reword “configurations” and I cannot see two conformations of plugs from Fig. 1b. Please show the details if authors want to highlight in the text.

A: Sorry for the mistake, and we have deleted this sentence. The two conformations of six gp16 subunits were indicated in the description part on the baseplate, and shown in Fig. 5.

The organization of the description part on neck is really confused and please re-write this part properly. I recommend to describe the structures following the order from top to bottom (portal, neck, and then adaptor).

A: We have revised the description part on the neck according to your suggestion.

Line 124-126: authors observed “two extra densities” and assigned them to “two ends of the dsDNA genome”. Please highlight two extra densities in Supplementary Fig. 1 a and do authors use “two ends of the dsDNA” to mean 5'- and 3'- end of dsDNA? If that case, how do authors know without determining the overall structure of viral genome? Please address it.

A: We have revised the Supplementary Fig. 1a (now Supplementary Fig. 2a in the revised manuscript) according to your suggestion. We could not assign the two densities

as two ends of the dsDNA without determining the overall structure of viral genome.

We have revised the related description.

Line 132: “polymerize” generally means that the proteins assemble into super-structure, like capsid or filament. Please reword it.

A: Revised.

Line 139-140: “Considering that all structure-known portal-adaptor complex are 12-fold symmetric” please cite the corresponding papers.

A: Added.

Line 141: “but failed in obtaining a best three-dimensional classification.” Change “best” to “reasonable” and did author try 3D classification here? If not, why use “classification”, not “reconstruction”?

A: Revised. Yes, we tried 3D classification here to search for a reasonable one for the further reconstruction of atomic models.

Line 144-146: authors mentioned that there is not fixed orientation between portal and adaptor. Could authors validate with detailed analysis? For example, authors could calculate minimal absolute different in azimuth angle based on the orientations determined from portal and adaptor.

A: Thanks for your suggestion. We tried to determine the interface between the

neck/gp5 and portal/gp2 in our cryo-EM map via rotating the structures and calculating the clashscores, but failed in obtaining a pose with the lowest clashscore, which represents the most favored relative orientation. It suggested that the difference between the relative rotations of 30° and 24° (for the 12-fold and 15-fold symmetric gp2 and gp5, respectively) is not distinguishable at the present resolution; however, the two components are docked to each other via an interface which is generally complementary in electrostatic potential and shape.

Line 147-148: “The neck is assembled on the adaptor, which coordinates the interactions with the portal, the connector and neck fibers.” The sentence is confused, because authors mentioned that portal, adaptor, and connector assemble to the neck (Line 102-103). Then how neck is assembled on the adaptor?

A: Revised.

Line 150: “whereas three adaptor subunits tightly interact with the trimeric gp82N via the interface between Loop α 1- α 2 and Loop α 5- α 6.” How “tightly” interactions? Are there some salt bridges? Please show the details on Fig. 2. In addition, please mention clearly the Loop α 1- α 2 of which protein interacts with the Loop β 5- β 6 of which protein.

A: According to your suggestion, we have added the detailed interaction as Supplementary Fig. 3b. Three gp5 subunits form a buried interface area of $\sim 1100 \text{ \AA}^2$ with the trimeric gp82N, mainly via seven pairs of hydrogen bonds.

Line 165-166: “gp80 and gp81, which are encoded in a same operon with gp82, constitute the neck fiber.” If I understand correctly, the neck fiber includes gp80, gp81, and gp82, not only gp80 and gp81. The sentence is confused and please reword it.

A: Revised.

Line 169: “As shown in the electrophoresis gel” Authors here showed the gel of the purified gp82-gp81-gp80 complex, but did not show the gel of the purified phage to confirm the stoichiometric ratios.

A: Several structural proteins, such as the wedge gp32, the tube gp10 and the terminator gp8, possess a molecular weight similar to gp80, gp81 and gp82, which make it difficult to distinguish these proteins on the gel. In addition, the occupancy of gp82-gp81-gp80 in the whole phage is not that high. Thus, the gel of the purified A-1(L) particles could not be applied to determine the stoichiometric ratios of three components of the neck fiber.

Line 176: “As expected, gp82 indeed possesses two separate domains gp82N and gp82C” What did authors expect? Please clarify or rewrite the sentence.

Line 181-183: “the modelled structure of gp80 trimer of relatively larger size should be localized to the most distal of the neck fiber.” How did author make the conclusion of the location of protein domain only based on the size? Please rewrite the sentence.

A: These two points have been revised.

Line 199-200: I am curious whether the structure of TMP follows the helical arrangement of tube and whether they have some contacts or not.

A: No, the TMP adopts a helical bundle structure with six subunits, all of which adopt a long helix structure; whereas the tube adopts a six-start helix structure. The TMP is inserted in the tube; however, it is hard to see direct contacts at the present resolution.

Line 201: please show the detailed interactions between TMP and baseplate in Supplementary Fig. 4a.

A: The detailed interactions were displayed as the Supplementary Fig. 8a in the revised manuscript.

Line 205-206: I can not see the β -hairpin of inner tube protruding towards the consecutive inner tube ring in Supplementary Fig. 4b. Please re-prepare the figure.

Line 226-227: It should be “disrupts” and “terminates”.

Line 244: please change to “name the three domains as lysozyme, endoglucanase, and peptidase domain, respectively”.

A: All points have been revised.

Line 248: “either yield much lower expression level of the recombinant protein” Fig. 5c-d do not show the corresponding data. Please provide the details.

A: Added as the Supplementary Fig. 8g-h.

Line 262-263: it's not clear to see the interactions between the N-terminal helical bundle of wedge and the LysM domain of the plug. Please re-prepare the proper figure to show the details.

A: Added as the Supplementary Fig. 9g.

Line 280-283: authors described the structure of STF. However, I did not see any descriptions about the contacts between STF and sheath (also from figure). Therefore, I am not sure that STF attaches on the sheath as authors stated in text. Please address it and tune down the statements about the unique features of tail fibers in A-1(L) compared to other phages.

A: Sorry for the unclear description. The STFs do not directly interact with the sheath as shown in the corresponding figures. We have revised the related description according to your suggestion.

Line 289-291: authors performed in vitro binding assays using the purified full-length LTF and STF, and showed that both proteins could bind to the cell surface. However, I suggest that authors should use the arm domain of LTF and the CBD domain of STF instead of full-length proteins.

A: According to your suggestion, the arm domain of LTF and the CBD domain of STF were purified and applied to in vitro binding assays. The results showed that the arm domain of LTF, but not the CBD domain of STF, binds to the cell surface of *Anabaena* sp. PCC 7120 (Fig. 6d, Supplementary Fig. 10b). In contrast, the stem-CBD domain of

STF could bind to the host cell surface (Fig. 6d). It suggested that the stem domain of STF is also required for the absorption of A-1(L) to the host.

Discussion part:

Line 334-335: please show the data or cite the reference about the statement “the distal CBM module in gp80 of the neck fiber does not possess binding activity...”.

A: Added as the Supplementary Fig. 10d.

Line 348: change “versicles” to “vesicles”.

Line 352-353: please cite the PDB entry and the corresponding reference when using the structure for analysis.

Line 360: I did not see multiple functions of the tail from the manuscript.

A: All points have been revised.

Line 363-364: authors state that A-1(L) could be an ideal cyanophage for the applications in synthetic biology. Is it available for genetic manipulation or re-engineering in A-1(L) phage? Since some phage systems could be re-engineered to target different hosts, what’s the advantage of A-1(L) compared to the established other phages?

A: Compared to the multiple strategies on bacteriophage engineering, no genome editing method has been reported for the cyanophage to date. As *Anabaena* sp. PCC 7120 is a genetic tractable model cyanobacterium, it is possible to manipulate the

genome of A-1(L) via the efficient genetic tools that have been developed in the host strain. In fact, we are now trying to establish the genetic manipulation method for A-1(L). Our structural studies would facilitate the engineering of A-1(L) as a chassis cyanophage for the future applications in synthetic biology.

Figure 1:

- a. I recommend to combine the two DNA organization schemes into one big panel.
- b. I highly recommend to re-prepare the figure with proper color annotation. I do not see any points why two different colors are applied to sheath and tube. Moreover, some positions of structural proteins mentioned in text are not clear (e.g., tube initiator, sheath initiator), please highlight by changing colors or showing the details with zoom-in figures.

A: We have revised the figure according to your suggestion. We adjusted the longitudinal cut view of A-1(L) in Fig. 1b to clearly show the position of structural proteins of the baseplate. In addition, a zoom-in view of the baseplate was shown as Fig. 5a.

Figure 2:

- d-f. The labels are overlapped with figure. Please fix it. Moreover, it's hard to distinguish different protomers from figure (e.g., three gp82 proteins bind to adaptor, but they look like one protein from figure). Please re-prepare figures with different colors on each protomers.

Figure 3:

a. It's difficult to see five neck fibers protruding from one phage. Please highlight it.

d. it's difficult to understand how authors deduce the molecular weight from the plot. It would be better to indicate the predicted molecular weight from the peak via dashed line.

Figure 4:

I highly recommend authors to re-prepare the figure and show the overall structures using atomic model instead of segmented map. In addition, I still did not see points why authors use two different colors for sheath and tube. For panel a, I cannot see clearly the sheath S2 N-tail because the color is too similar to tube.

A: All points have been revised.

Figure 5:

f-h. I cannot see the interaction details. Please highlight by showing zoom-in structures, not just remarked with red circles.

A: The interaction details have been added in the Supplementary Fig. 9.

Figure 6:

e. It's hard to distinguish different domains of STF. Please color the STF based on the domains shown in panel c.

Supplementary Figure 2:

authors stated that the electrostatic potentials on each neck elements are complementary.

However, I cannot see the interface boundary between each neck components. Please highlight them in the figure.

Supplementary Figure 3:

please re-organize the panels. For panel c, I would like to show the diameter difference between gp80 and gp81, instead of showing structural superposition.

Supplementary Figure 4:

c. please cite the corresponding PDB entries.

Supplementary Figure 5:

e: all domain annotations miss “domain”.

A: All above points have been revised.

Reviewer #1 (Remarks to the Author):

The revised version of the MS "Structure of the intact tail machine of *Anabaena myophage A-1(L)*" reads significantly better than the previous version. The authors addressed the comments of the referees adequately and performed additional experiments. It wasn't this reviewer's intent to ask for additional experiments for this paper - it was a suggestion for future work. These new experiments make the paper much stronger and more interesting from the functional point of view.

I have only one major concern that should have been addressed in the previous revision and a few minor comments.

My only major concern is about the presentation of experimental data that describe protein complexes and binding assays. Fig. 3f and 3g show the structure and assembly of the neck fiber. An SDS PAGE of the sample shown in both EMs is required. The chromatograms do not tell us much about what is exactly imaged in the EMs. SDS-PAGE does. For the same reason, all images showing binding assays of fluorescently labeled proteins must be accompanied by SDS-PAGEs of samples that were used in these assays (Fig. 6d, Suppl. Fig. 10b, c, d). Without SDS-PAGEs, the quality of the sample used for labeling is unknown.

Lines 321-333 state that the CBD of the STF does not bind to host cells while a longer fragment that also contains the stem region does. We do not know anything about these proteins. Are they trimeric and folded correctly? Presumably, the longer construct is indeed trimeric as it contains the copied coil "stem" region, a powerful "trimerizer". Whether the CBD without the stem is trimeric and stable is a question to which we have no answer. In summary, all binding assays must be accompanied by a more detailed characterization of proteins used in these assays.

Minor comments.

The Abstract is too detailed to a point that it is difficult to read and understand. The language is also awkward: "Encircling the helical bundle of tape measure proteins, the 1045-Å-long contractile tail is composed..." The tape measure protein is a tail component. The whole cannot "encircle" a component. Basically, I strongly advise discussing the Abstract with a non-expert in the field and rephrasing things to become understandable for a non-expert.

L. 302. "differ a lot". "A lot" is not something that should be used in a scientific article.

L. 399-402. "Both the LTFs and STFs function as RBPs to bind to the cell surface of host *Anabaena* sp. PCC 7120, and might recognize various host receptors. It is consistent with a previous report that the LTF, but not the STF, specifically targets the O antigen of lipopolysaccharides."

Sorry, I do not follow the logic here. What is "consistent"? You show here, in this paper, that both STF and LTF bind *Anabaena* sp. PCC 7120!

Suppl. Fig. 7c. Labels for T4 and Pam3 proteins have been switched.

Suppl. Fig 8b. The three domains of gp15 are marked with differently colored labels, but not colored in different colors. Correct.

Reviewer #2 (Remarks to the Author):

The quality of manuscript "Structure of the intact tail machine of *Anabaena myophage A-1(L)*" is improved a lot after revision, and most of my concerns in last peer-review has been properly addressed. However, I still have some minor concerns that authors should revise.

1. Authors revised the part of tail fiber and tuned down the statements, showing that the STFs fold back towards the sheath (but not pack/stick on the sheath). However, some sentences in main text are still kept as original and are inconsistent with current version.

Line 33 in Abstract: "which pack against each other pairwise, are folded back and finally **stick on the sheath.**"

Line 84 "both the long and short tail fibers are folded back, and **pack against** the sheath pairwise."

Line 117-118 "and six short tail fibers (STFs, gp33 trimers), are folded back and **pack against** the sheath pairwise."

2. The table S1 is messed up and could not be understood. Please correct it.

3. The format of reference is not consistent. Please re-format this part.

4. Some statements in the discussion need to be revised.

Authors stated that both tail fibers might involve in the target recognition (Line 400-401), but then stated "It is consistent with a previous report that the LTF, but not the STF, specifically targets the O antigen of lipopolysaccharides." (Line 401-402). I do not see that it is consistent with the conclusion from authors.

Line 409-412: authors compared the central spike of A-1(L) and the tCIS in cyanobacteria, then made a conclusion that A-1(L) might use a similar penetration mechanism. However, the mechanism and the target of tCIS is still unclear.

Reviewer #1:

The revised version of the MS “Structure of the intact tail machine of *Anabaena* myophage A-1(L)” reads significantly better than the previous version. The authors addressed the comments of the referees adequately and performed additional experiments. It wasn't this reviewer intent to ask for additional experiments for this paper - it was a suggestion for future work. These new experiments make the paper much stronger and more interesting from the functional point of view.

I have only one major concern that should have been addressed in the previous revision and a few minor comments.

Q1. My only major concern is about the presentation of experimental data that describe protein complexes and binding assays. Fig. 3f and 3g show the structure and assembly of the neck fiber. An SDS-PAGE of the sample shown in both EMs is required. The chromatograms do not tell us much about what is exactly imaged in the EMs. SDS-PAGE does. For the same reason, all images showing binding assay of fluorescently labeled proteins must be accompanied by SDS-PAGEs of samples that were used in these assays (Fig. 6d, Suppl. Fig. 10b, c, d). Without SDS-PAGEs, the quality of the sample used for labeling is unknown.

A: The corresponding SDS-PAGE profiles have been added as Supplementary Figs. 4d, 8g-h and 11 according to your suggestion.

Q2. Lines 321-333 state that the CBD of the STF does not bind to host cells while a

longer fragment that also contains the stem region does. We do not know anything about these proteins. Are they trimeric and folded correctly? Presumably, the longer construct is indeed trimeric as it contains the copied coil “stem” region, a powerful “trimerizer”. Whether the CBD without the stem is trimeric and stable is a question to which we have no answer. In summary, all binding assays must be accompanied by a more detailed characterization of proteins used in these assays.

A: Thanks for your suggestion. Gel filtration chromatography and SDS-PAGE profiles of proteins used in all the binding assays have been added as Supplementary Fig. 11. The chromatography profiles indicated that both the CBD and stem-CBD of gp33 are trimeric and folded correctly, in agreement with our structure of trimeric gp33 that also contains trimeric interfaces among the CBDs.

Minor comments.

Q3. The Abstract is too detailed to a point that it is difficult to read and understand. The language is also awkward: “Encircling the helical bundle of tape measure proteins, the 1045-Å-long contractile tail is composed...” The tape measure protein is a tail component. The whole cannot “encircle” a component. Basically, I strongly advise discussing the Abstract with a non-expert in the field and rephrasing things become understandable for a non-expert.

A: Revised.

Q4. L. 302. “differ a lot”. “A lot” is not something that should be used in a scientific

article.

A: Revised.

Q5. L. 399-402. “Both the LTFs and STFs function as RBPs to bind to the cell surface of host *Anabaena* sp. PCC 7120, and might recognize various host receptors. It is consistent with a previous report that the LTF, but not the STF, specifically targets the O antigen of lipopolysaccharides.”

Sorry, I do not follow the logic here. What is “consistent”? You show here, in this paper, that both STF and LTF bind *Anabaena* sp. PCC 7120!

A: Revised.

Q6. Suppl. Fig. 7c. Labels for T4 and Pam3 proteins have been switched.

A: Revised.

Q7. Suppl. Fig 8b. The three domains of gp15 are marked with differently colored labels, but not colored in different colors. Correct.

A: Revised.

Reviewer #2:

The quality of manuscript “Structure of the intact tail machine of *Anabaena* myophage A-1(L)” is improved a lot after revision, and most of my concerns in last

peer-review has been properly addressed. However, I still have some minor concerns that authors should revise.

Q1. Authors revised the part of tail fiber and tuned down the statements, showing that the STFs fold back towards the sheath (but not pack/stick on the sheath). However, some sentences in main text are still kept as original and are inconsistent with current version.

Line 33 in Abstract: “which pack against each other pairwise, are folded back and finally **stick on the sheath.**”

Line 84 “both the long and short tail fibers are folded back, and **pack against** the sheath pairwise.”

Line 117-118 “and six short tail fibers (STFs, gp33 trimers), are folded back and **pack against** the sheath pairwise.”

A: Revised.

Q2. The table S1 is messed up and could not be understood. Please correct it.

A: Revised.

Q3. The format of reference is not consistent. Please re-format this part.

A: Revised.

Q4. Some statements in the discussion need to be revised.

Authors stated that both tail fibers might involve in the target recognition (Line 400-401), but then stated “It is consistent with a previous report that the LTF, but not the STF, specifically targets the O antigen of lipopolysaccharides.” (Line 401-402). I do not see that it is consistent with the conclusion from authors.

Line 409-412: authors compared the central spike of A-1(L) and the tCIS in cyanobacteria, then made a conclusion that A-1(L) might use a similar penetration mechanism. However, the mechanism and the target of tCIS is still unclear.

A: Revised.

Reviewer #1 (Remarks to the Author):

I applaud the authors for providing additional data and description of their experiments and for revising the text according to reviewers' suggestions.

I have only one small comment regarding the way the interaction of the two RBPs with their receptors is described. The work presented here shows that both RBPs of A-1(L) can simultaneously bind to cells. This means they interact with different cell surface receptors. Not various as stated in the text, but different.

Lines 330-331. "which might recognize and bind to various receptors on the host cell surface."
should be changed to

"which recognize and bind to different receptors on the host cell surface."

Line 402. "might recognize various host receptors"

should be changed to

"recognize different host receptors"

In both cases, "might" should be omitted, as the results are rather definitive.

The fit of "8KE9_gp33 CBD.pdb" to "gp33_EMD-37150.mrc" is very poor. See the image below. This model needs to be refitted into the map, not just a bit of re-refined.

The baseplate model and map are both good, except for the tip of the central spike protein. See the screenshot below. Even the chain trace is messed up in this region because the density is too poor to allow for model building. The authors must perform focused refinement of that part to improve the map or delete the atomic model of the very tip. Notice how the map quality drops when the structure gets pointier.

"8KEC_tail fiber.pdb" fit to "tail_fiber_EMD-37152.mrc" is good. One or two histidine side chains need to be flipped where they form metal-binding histidine cages (HxH motif times three in the trimer). But that's a huge structure, so one or two little errors are fine.

"8KEE_sheath-tube.pdb" fit to "sheath_tube_EMD-37153.mrc" is good except for the C-terminus of the tube protein (see the image below). The C-term of the sheath is also poorly defined, so the authors might want to not build the very C-terminus. The tube protein crashing with the sheath (as in the image below) is really not good! Something like this should be in the PDB.

ut Ligand Refine Morph

Sphere Refine + Tandem Refine Undo Molecule

"8KEF_Neck.pdb" fit to "neck_EMD-37154.mrc" is good.

"8KEG_adaptor-neck fiber.pdb" fit to "adaptor_neck_fiber_EMD-37155.mrc" is good. **This is one crazy structure!** But **correct**, as far as I can tell.

"8TS6_portal.pdb" and "portal_EMD-41590.mrc" are good, except for the loop shown in the image below. There is no density for it, so it should not be built. There is another protruding loop for which the density is poor. It can probably be kept.

Response to the cryo-EM feedback

Q1. The fit of “8KE9_gp33 CBD.pdb” to “gp33_EMD-37150.mrc” is very poor. See the image below. This model needs to be refitted into the map, not just a bit of re-refined.

A: The model has been finely refitted into the map.

Q2. The baseplate model and map are both good, except for the tip of the central spike protein. See the screenshot below. Even the chain trace is messed up in this region because the density is too poor to allow for model building. The authors must perform focused refinement of that part to improve the map or delete the atomic model of the very tip. Notice how the map quality drops when the structure gets pointier.

A: According to your suggestion, we performed the focused refinement against the tip of the central spike protein. However, the map quality was not largely improved (left), and the density is poor to only allow the model building of main chains of this part (middle). So, we deleted the residues Thr246~Gly269 (right) at the very tip of central spike in the final model.

left

middle

right

Q3. “8KEC_tail fiber.pdb” fit to “tail_fiber_EMD-37152.mrc” is good. One or two histidine side chains need to be flipped where they form metal-binding histidine cages (HxH motif times three in the trimer). But that’s a huge structure, so one or two little errors are fine.

A: Corrected.

Q4. “8KEE_sheath-tube.pdb” fit to “sheath_tube_EMD-37153.mrc” is good except for the C-terminus of the tube protein (see the image below). The C-term of the sheath is also poorly defined, so the authors might want to not build the very C-terminus. The tube protein crashing with the sheath (as in the image below) is really not good! Something like this should be in the PDB.

A: To avoid clashes, we have deleted the C-terminus (residues Gln163~Phe167) of the tube protein (top). Moreover, we have refined the C-terminus of the sheath protein (bottom), and deleted the last C-terminal residue Val506.

top

bottom

“8KEF_Neck.pdb” fit to “neck_EMD-37154.mrc” is good.

“8KEG_adaptor-neck fiber.pdb” fit to “adaptor_neck_fiber_EMD-37155.mrc” is good.

This is one crazy structure! But correct, as far as I can tell.

Q5. "8TS6_portal.pdb" and "portal_EMD-41590.mrc" are good, except for the loop shown in the image below. There is no density for it, so it should not be built. There is another protruding loop for which the density is poor. It can probably be kept.

A: According to your suggestion, we have deleted the residues Ser137~Thr155 in each subunit of the portal.

Response to the Reviewer comments.

Reviewer #1:

I applaud the authors for providing additional data and description of their experiments and for revising the text according to reviewers' suggestions.

I have only one small comment regarding the way the interaction of the two RBPs with their receptors is described. The work presented here shows that both RBPs of A-1(L) can simultaneously bind to cells. This means they interact with different cell surface receptors. Not various as stated in the text, but different.

Lines 330-331. "which might recognize and bind to various receptors on the host cell surface."

should be changed to

“which recognize and bind to different receptors on the host cell surface.”

Line 402. “might recognize various host receptors”

should be changed to

“recognize different host receptors”

In both cases, “might” should be omitted, as the results are rather definitive.

A: Thanks for your suggestion. We have revised the corresponding sentences in the text.